# Microstructure-based modelling of snow mechanics: experimental evaluation on the cone penetration test

Clémence Herny[1,2], Pascal Hagenmuller[1], Guillaume Chambon[2], Isabel Peinke[1], Jacques Roulle[1]

[1]Univ. Grenoble Alpes, Université de Toulouse, Météo-France, CNRS, CNRM, Centre d'Etude de la Neige, Grenoble, France
[2]Univ. Grenoble Alpes, CNRS, INRAE, IRD, Grenoble INP, IGE, Grenoble, France

*Correspondence to*: Clémence Herny (clemence.herny@gmail.com)

**Abstract.** Snow is a complex porous material presenting a variety of microstructural patterns. This microstructure largely controls the mechanical properties of snow, although the relation between the micro and macro properties remains to be better understood. Recent developments based on the discrete element method (DEM) and three-dimensional microtomographic data make it possible to reproduce numerically the brittle mechanical behaviour of snow. However, these developments lack experimental evaluation so far. In this study, we evaluate a DEM numerical model by reproducing cone penetration tests on centimetric snow samples. The microstructures of different natural snow samples were captured with X-ray microtomography before and after the cone penetration test, from which the grain displacements induced by the cone could be inferred. The tests were conducted with a modified Snow MicroPenetrometer (5 mm cone diameter), which recorded the force profile at a high resolution. In the numerical model, an elastic brittle cohesive contact law between snow grains was used to represent the cohesive bonds. The initial positions of the grains and their contacts were directly derived from the tomographic images. The numerical model was evaluated by comparing the measured force profiles and the grain displacement fields. Overall, the model satisfactorily reproduced the force profiles in terms of mean macroscopic force (mean relative error of about 20%) and the amplitude of force fluctuations (mean relative error of about 55%), while the correlation length of force fluctuations was more difficult to reproduce (mean relative error of about 40% for two samples out of four and by a factor $\geq 8$ for the other two). These characteristics were, as expected, highly dependent on the tested sample microstructure, but they were also sensitive to the choice of the micro-mechanical parameters describing the contact law. A scaling law was proposed between the mechanical parameters, the initial microstructure characteristics and the mean macroscopic force obtained with the DEM numerical model. The model could also reproduce the measured deformation around the cone tip (mean grain displacement relative error of 57% along the horizontal axis), with a smaller sensitivity to the contact law parametrisation in this case. These detailed comparisons between numerical and experimental results give confidence in the reliability of the numerical modelling strategy and opens promising prospects to improve the understanding of snow mechanical behaviour.

## 1 Introduction

Snow is a brittle and porous material existing on Earth close to its melting point. The thermodynamical conditions in the clouds govern the snowflake morphology and, once deposited on the ground, snow continues to evolve via metamorphism. The snow material is thus characterised by a large variety of microstructural patterns (grain size, grain shape, density) classified into different snow types (Fierz et al., 2009). It has been established that the snow microstructure controls the properties of snow (Shapiro et al., 1997; Johnson and Schneebeli, 1999; Schneebeli, 2004). For instance, weak layers involved in avalanche triggering (Schweizer et al., 2003) are usually constituted of specific snow types (depth hoar, surface hoar, precipitation particle, faceted crystals) characterised by low cohesion and low strength (Jamieson and Johnston, 1992). The link between the snow microstructure and its properties, especially its mechanical properties, is still not well understood, even if it is crucial for many applications, such as avalanche forecasting (Schweizer et al., 2003, Jamieson and Johnston, 1992), snowpack modelling (Calonne et al. 2014), ice core interpretation (Montagnat et al. 2020) or geotechnics (Shapiro et al., 1997). In particular, the brittle failure occurring at high shear rates ($> 10^{-4}$ s$^{-1}$) during the release of an avalanche remains represented by very coarse empirical laws (Brun et al., 1992; Bartelt, et al. 2002; Vionnet et al. 2012). In this elastic-brittle regime (rapid and large deformations), the mechanical behaviour of snow is thought to be mainly controlled by bond failures and grain rearrangements (Narita, 1983).

The snow microstructure and its evolution can be captured at high resolution (typically 10-50 µm) with X-ray micro tomography imaging (µCT) (Coléou et al., 2001; Freitag et al., 2004; Schneebeli, 2004; Heggli et al., 2011). This non-destructive method preserves the snow microstructure and resolves the shape of snow grains, grain bonds and porosity which is of primary importance for mechanical studies. In particular structural properties of snow, such as density, specific surface area (SSA), correlation length, bond characteristics, can be evaluated from tomographic data (e.g. Schneebeli, 2004; Schneebeli et al., 2004; Hagenmuller et al., 2014a; Calonne et al., 2014; Proksch et al., 2015). The tomographic data are also used as a basis for numerical modelling (Schneebeli, 2004; Schneebeli et al., 2004; Hagenmuller et al., 2015) or calibration/validation data of statistical empirical models retrieving grain-scale physical and mechanical properties from other measurements (e.g. Proksch et al., 2015; Reuter et al., 2019). However, tomographic imaging is time-expensive and not adapted to routine measurements in the field.

The mechanical properties of snow are commonly derived from Cone Penetration Test (CPT) measurements, which is an objective and relatively easy-to-set-up method (Schneebeli and Johnson, 1998). This method has been widely used to characterise soil stratigraphy (Lunne et al., 1997) and adapted to snowpack stratigraphy (Gubler, 1975; Schaap and Fohn, 1987; Dowd and Brown, 1986; Schneebeli and Johnson, 1998; Mackenzie and Payten, 2002; McCallum, 2014). The CPT provides a force profile by measuring the resisting force exerted on a conic tip penetrating, at a constant rate, into a material. The development of high-resolution digital penetrometers dedicated to snow studies (Schneebeli and Johnson, 1998; Mackenzie and Payten, 2002; McCallum, 2014) has provided the possibility to resolve the force profile at a microscopic scale

and capture the high-frequency fluctuations of the force signal up to a metre depth. Such force penetration profiles contain valuable information on the snow structural parameters at macro- and micro-scale (Löwe and van Herwijnen, 2012).

Interpretation of the CPT requires a good understanding of the interactions between the cone tip and the snow grains. Several studies aimed to investigate the grain displacement field around the tip. Particle Image Velocimetry (PIV) imaging was performed to quantify the 2D displacement field of snow grains while the tip penetrates into the material (Floyer and Jamieson, 2010; Herwijnen, 2013; LeBaron et al., 2014). Peinke et al. (2020) developed a grain tracking algorithm to reconstruct from μCT the 3D displacement field of snow grains induced by a CPT. All these studies revealed the development of a compaction zone (CZ) in front of the tip.

Various mechanical or statistical models have been developed to interpret the CPT penetration signal in terms of mechanical properties. The cavity expansion model (CEM) (Bishop et al., 1945; Yu and Carter, 2002) has been applied to snow by Ruiz et al. (2016) and Peinke et al. (2020). This model considers snow as a continuum and describes the elastic-plastic deformation of the material around the tip in order to retrieve macroscopic material properties (cohesion, friction, etc.). The continuum assumption becomes invalid for a ratio between cone diameter and mean grain diameter lower than 20 typically (Bolton et al. 1993), leading to potentially erroneous interpretations of the CPT results. Alternatively, the shot noise model interprets the force signal and its fluctuations as a superposition of independent elastic–brittle ruptures occurring next to the tip (Schneebeli and Johnson, 1999; Marshall and Johnson, 2009; Löwe and van Herwijnen, 2012) and retrieves microstructural properties (bond rupture force, etc.) The penetration process is generally modelled as a Homogeneous Poisson Process (HPP) with a constant intensity (Löwe and van Herwijnen, 2012). Peinke et al. (2019) have generalised the HPP method to account for the transient phase of the penetration process, attributed to the development of the CZ (Peinke et al., 2019). These authors used a Non-Homogeneous Poisson Process (NHPP) considering a depth dependency of the intensity (number of bond failures per penetration increment). Yet, the assumption of independent elastic-brittle rupture events essentially neglects the development of a CZ (Johnson and Schneebeli, 1999; Schneebeli, 2001; Herwijnen, 2013; LeBaron et al., 2014; Ruiz et al. 2017). Therefore, none of these two models appear to fully account for the specificity of snow deformation induced by CPT. Additional investigations are required to better understand the tip interaction with snow and better interpret the force measurements.

Recently, numerical approaches have been developed to study the mechanical response of snow by explicitly accounting for the microstructure (Johnson and Hopkins, 2005; Gaume et al., 2015, 2017; Hagenmuller et al., 2015; Wautier et al., 2015; Mede et al. 2018b, 2020; Bobillier et al., 2020, 2021). Snow is described as a granular material and modelled by the discrete element method (DEM) in a high shear rate regime. The complexity of the snow microstructure can be taken into account by feeding the DEM simulations with high-resolution 3D reconstructions obtained with μCT. These simulations have provided new insights into the snow mechanical behaviour, such as the dependence of snow strength to microstructure properties (Hagenmuller et al., 2015) or the identification of different failure modes in shear loading (Mede et al., 2018b, 2020). The downside of this method is that it is time-consuming, and simulations can only be performed on small samples (up to a few centimetres). Furthermore, these numerical models still lack direct experimental evaluation.

In this context, the aim of this study was to evaluate a microstructure-based DEM model using recent CPT experimental data
performed in a controlled environment (Peinke et al., 2020). The dataset includes µCT images of the samples acquired before
and after the tests. The deformation induced by the CPT (strain rate of about $10^2$ s$^{-1}$, Reuter et al., 2019) belongs to the elastic-
brittle regime (Narita, 1983; Floyer and Jamieson, 2010) and is therefore suitable for DEM simulation. The results of the
numerical model are directly compared to experimental data in terms of (1) macroscopic force profile and associated statistical
indicators and (2) grain displacements induced by the cone penetration. A systematic sensitivity analysis to DEM mechanical
parameters, including Young's modulus, cohesion and friction coefficient, was performed to find the combinations of
parameters that best reproduce experimental results. Finally, the role of the microstructure was also investigated by performing
DEM simulations for different snow types. The evaluation of the numerical model provides the opportunity to better understand
the mechanisms at play during snow deformation in an elastic-brittle regime and better interpret CPT profiles.
We first present the experimental dataset and the numerical methods. The data processing used to compare experimental and
numerical results is also explained. The results of the DEM, the sensitivity analysis to mechanical parameters and the
comparison to experimental results are then presented. The relevance of the DEM model and the limits of our approach are
eventually discussed before concluding.
**2 Methods**
**2.1 Experimental measurements**
The experimental dataset used in this study has been acquired by Peinke et al. (2020) and is only briefly presented in this paper.
The methodology comprises collection and preparation of snow samples, acquisition of high-resolution micro-tomographic
images and cone penetration tests (CPT).
**2.1.1 Snow sample preparation**
Blocks of natural snow were sampled in the French Alps near Grenoble and stored at -20°C in a cold room. The materials
collected were representative of the variety of seasonal snow types (Table 1), namely rounded grains (RG), large rounded
grains (RGlr), depth hoar (DH) and precipitation particles (PP), with distinct bulk densities and specific surface areas (SSA).
The samples were then prepared in a cold room at -10°C by sieving the different snow types into aluminium cylinders of 20
mm height and 20 mm diameter. All samples were prepared at least 24 hours before the measurements in order for the bonds
between grains to rebuild after sieving.
**2.1.2 Micro-Tomography (µCT)**
Tomographic scans of each sample were acquired before and after performing the CPT to capture the initial and final
microstructure of the snow, respectively. An X-ray tomograph (DeskTom130, RX Solutions) operating at a pixel size of 15
µm pix$^{-1}$, a voltage of 80 kV and a current of 100 µA was used. During tomographic scanning, the samples were maintained
at a constant and uniform temperature of -10°C in a cryogenic cell (CellDyM, Calonne et al. (2015)). Each scan, consisting of
1440 2D radiographs, was reconstructed to obtain 3D grayscale images representing the attenuation coefficients of the different
materials composing the samples. The grayscale images were then transformed into binary (ice matrix – pore space) segmented
images using an energy-based segmentation algorithm (Hagenmuller et al., 2013).

**2.1.3 Cone Penetration Test (CPT)**

After the initial micro-tomography scan, a CPT was performed on the snow samples using a modified SnowMicroPenetrometer
(SMP version 4, Schneebeli and Johnson, 1998). The specific rod used by Peinke et al. (2020) displays a conic tip with an
apex angle $a$ of 60° and a maximum cone radius equal to the rod radius $R$ of 2.5 mm. The rod was inserted vertically into the
snow sample at a constant penetration speed $v$ of 20 mm s$^{-1}$. The resisting force applied on the penetrometer (cone and rod)
was recorded at every 4 μm of penetration increment (i.e., 5 kHz frequency). The SMP sensor (Kistler sensor type 9207) can
measure forces up to 40 N with a resolution of 0.01 N. The tip was stopped at depths between 7 and 15 mm, i.e., 5 to 13 mm
above the sample bottom, to avoid boundary effects (Peinke et al., 2020). The experimental force profiles are presented in
Figure S26.

**2.2 Numerical modelling**

Snow is here considered as a granular cohesive material. The high strain rate ($> 10^{-4}$ s$^{-1}$) induced by the tip penetration in the
snow sample is considered to lead to brittle deformations, with inter-granular damage and grain rearrangements (Narita, 1983;
Johnson and Hopkins 2005; Hagenmuller et al., 2015). We adopted an approach based on DEM to simulate the cone penetration
tests in the measured snow samples. The mechanical model, based on YADE software (Šmilauer et al., 2015), is adapted from
the work of Hagenmuller et al. (2015) and Mede et al. (2018a, b and 2019).
The setting-up of the simulations involves different steps, namely the generation of the initial conditions based on measured
snow microstructures, the definition of the contact laws between the snow grains, and the setting of the boundary conditions
to reproduce the CPT configuration.

**2.2.1 Grain segmentation and grain shape representation**

The DEM model was fed by the 3D ice-air images derived from μCT. The continuous ice matrix was first segmented into
individual grains based on geometrical criteria, as described by Hagenmuller et al. (2013). The main idea of the approach is
to detect potential mechanical weakness zones (i.e., the bonds) based on the principal minimal curvature $\kappa_T$ and a contiguity
parameter $c_T$. The threshold on curvature $\kappa_T$ was set to 1.0 for RG, RGlr and DH samples and to 0.7 for PP sample; the contiguity
parameter was set to 0.1 for all the samples (see Hagenmuller et al., 2013 for details).
To construct the DEM sample, the irregular shape of the grains was approximated by filling the grain volume with a population
of overlapping spheres (Fig. 1). The position of these spheres was derived from the medial axis of the structure (Coeurjolly et
al., 2007; Mede et al., 2018a) and redundant spheres were discarded based on a power diagram filter (Coeurjolly et al., 2007).

This grain shape representation by a multitude of spheres preserves the capability of YADE to handle sphere-sphere contact detection. However, a high number of spheres slows down the simulations. We thus further decimated the number of spheres by approximating the grain shape. We only selected the spheres with a radius larger than a threshold $L$ (voxel) and with a relative coverage larger than $S$ (i.e., the ice volume associated with the sphere according to the power diagram should be larger than $S$ times the sphere volume (Coeurjolly et al., 2007). A trade-off must be found between this grain shape approximation, influencing the simulation accuracy, and the number of spheres influencing the numerical cost. Eventually, the spheres belonging to the same grain were clumped together in rigid aggregates constituting single discrete elements (DE). A detailed sensitivity analysis was conducted (see supplementary material, Table S1 and Fig. S1) to determine the optimal values of $L$ and $S$ parameters. Note that this grain shape approximation might also lead to delete the smallest grains in the numerical samples, as they cannot be covered with the chosen parameters $L$ and $S$. The grain number difference and shape approximation of the numerical sample compared to initial the segmented μCT image can be quantified by computing the volumetric error $E_V$. The final chosen $L$ and $S$ values for each snow type, with the associated volumetric $E_V$ and mechanical $E_M$ errors (defined in Sect. S1.1), can be found in Table 1.

| Sample name | Snow type | Sieve size (mm) | Bulk density (kg m$^{-3}$) | SSA (m$^2$ kg$^{-1}$) | L (vx) | S | Number of spheres | Number of grains | Number of initial cohesive interactions between grains | Initial contact density $\nu$ | $E_V$ (%) | $E_M$ (%) |
|---|---|---|---|---|---|---|---|---|---|---|---|---|
| RG | Rounded Grains | 1.6 | 289 | 23.0 | 5 | 0.3 | 514917 | 27560 | 47736 | 0.55 | 42.3 | 5.3 |
| RGlr | Large Rounded Grains | 1 | 530 | 10.1 | 5 | 0.3 | 270143 | 8488 | 24005 | 1.63 | 14.6 | 4.2 |
| DH | Depth Hoar | 1.6 | 364 | 15.9 | 5 | 0.2 | 743546 | 11211 | 24258 | 0.86 | 24.7 | 14.3 |
| PP | Precipitation Particle | 1.6 | 91.3 | 53.5 | 2 | 0.5 | 1797567 | 95022 | 125805 | 0.13 | 32.2 | 10.3 |

**Table 1: Overview of the snow samples analysed in this study and parameters of DEM grain shape representation. Sample names were given according to the snow type classification (Fierz et al., 2009). The sample density and specific surface area (SSA) were derived from the micro-tomographic images (Peinke et al., 2020). The initial contact density was computed according to Eq. 10. The minimum radius of the sphere $L$ and the minimum sphere coverage $S$ were determined through a sensitivity analysis presented in Sect. S1.1. The resulting number of spheres, grains and cohesive grain-grain interactions are indicated, as well as the volumetric error $E_V$ and the mechanical error $E_M$ associated with each grain shape representation.**

### 2.2.2 Interactions and contact law

The contacts between adjacent grains were identified during the grain segmentation phase. In the DEM simulations, each grain contact is represented by several sphere-sphere interactions. The interactions between spheres are described by an elastic brittle

cohesive contact law characterised by four parameters, namely the normal and the shear contact stiffness $K_N$ and $K_S$, the
adhesion $A$, and the friction angle $\varphi$. The normal force $F_N$ between two spheres is computed as proportional to the distance
between the two sphere surfaces $x_N$, and limited by the adhesion value in the tensile regime ($x_N > 0$):
$$F_N = K_N x_N \leq A. \tag{1}$$
The shear force $F_S$ is proportional to the shear displacement between the spheres $x_{S,}$, with a maximal value given by the sum
of adhesion and friction:
$$F_S = K_S x_S \leq A + F_N \tan(\varphi). \tag{2}$$
If the force exceeds the threshold, either in tension or in shear, the cohesive bond is broken. As long as the spheres remain in
contact after the bond is broken, friction remains active in shear. In the initial state, all interactions in the numerical sample
are considered cohesive. While the sample deforms, grain displacements lead to progressive breakage of the initial cohesive
interactions and the potential creation of new contacts. These new interactions are frictional only (no cohesion), meaning that
sintering mechanisms are not considered in this study.
The force of a given intergranular cohesive contact corresponds to the sum of all the associated sphere-sphere interactions.
Based on the total contact surface between two grains (obtained from the µCT image) and the number of associated sphere-
sphere interactions, each sphere-sphere interaction $i$ can be associated with a representative contact surface $D_i$. In order to
recover the correct cohesion strength between two grains, the adhesion parameter $A$ was defined for each sphere-sphere
interaction as:
$$A_i = D_i C, \tag{3}$$
with $C$ (Pa) the cohesion of ice. In YADE, by default, the contact stiffnesses are computed based on the radii of the spheres in
interaction and two elastic material parameters, namely the Young's modulus $E$ and the Poisson ratio $v$. For our computations,
to ensure that all cohesive sphere-sphere interactions between two grains break at the same separation distance, the computation
of the normal stiffness was redefined as:
$$K_{N,i} = \frac{D_i E}{r_{mean}}, \tag{4}$$
where $r_{mean}$ (m) is a characteristic length constant for all the interactions in the numerical sample, taken as the mean sphere
radius. The shear stiffness is then defined as:
$$K_S = v \times K_N. \tag{5}$$
Note that due to the rather arbitrary characteristic length considered in the definition of the normal stiffness [Eq. (4)], which
depends on the grain shape approximation, as well as to the simple linear relation considered for the normal force [Eq. (1)],
the contact-level YADE Young's modulus $E$ should not be regarded as the "true" Young's modulus of the material, but rather
as a representative parameter of the elastic properties at the contacts.

### 2.2.3 Simulation setup and critical time step

In order to evaluate the DEM model, we have implemented a CPT configuration similar to the experimental setup used by
Peinke et al. (2020) (Fig. 1). The snow sample is contained in a rectangular box open at the top. The box is about 12.4 mm
along the x- and y-axis and about 15 mm along the z-axis. The vertical and horizontal box sizes were reduced compared to the
20 mm height and 20 mm diameter respectively of the sample holder used by Peinke et al. (2020). This choice has been
motivated by (1) simplifying the geometry with a rectangular numerical sample, (2) matching the sample height imaged with
μCT and (3) reducing the computational time. A sample size sensitivity analysis has been performed to ensure that border
effects are not introduced by reducing the sample size (Fig. S2). The penetrometer tip displays a maximal radius $R$ of 2.5 mm
and an apex angle $a$ of 60°. Initially in a centered position at the box surface, it is displaced downwards through the sample at
a constant speed of 20 mm s$^{-1}$. The simulation stops when the tip reaches the bottom of the box. The walls (box and tip) are
represented by facets with rigid boundary conditions. The gravity is set to 9.81 m s$^{-2}$.

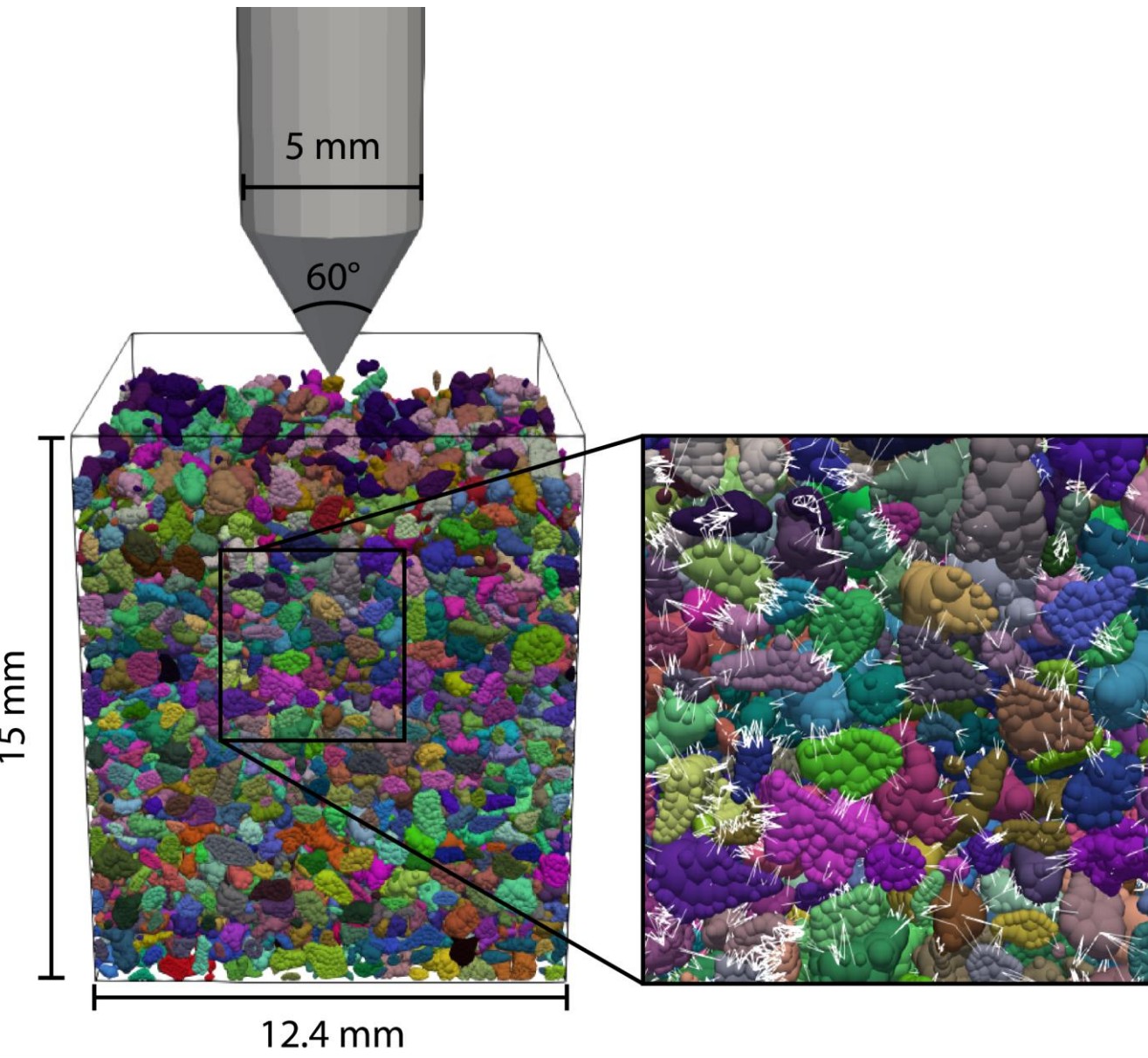

**Figure 1: Illustration of DEM CPT modelling for the RGlr sample. The penetrometer is moving downward at a constant speed of**
**20 mm s⁻¹. Snow grains (represented with different colours) are modelled by overlapping spheres clumped together. The zoomed**
**window shows the initial cohesive interactions between the spheres of adjacent grains (white lines).**

The stability of the explicit integration scheme is ensured by estimating the critical time step, based on the propagation speed
of elastic waves in the sample (Zhao, 2017):
$$\Delta t_{cr} = min \left( \frac{m_i}{K_{N,i}} \right)^{0.5} ,$$ (6)
with $m_i$ and $K_{N,i}$ the mass and normal stiffness of the DE $i$. The mass $m_i$ or, equivalently the material density $\rho$, can be artificially
increased to increase the time step (Hagenmuller et al., 2015). A numerical sensitivity analysis (Fig. S3) has shown that
increasing the density by a factor $f$ equal to 100 does not affect the simulation results, while significantly reducing the
computing time. Finally, a Cundall's non-viscous damping coefficient $\Lambda$ was applied to the particle acceleration to dissipate
kinetic energy and avoid numerical instabilities (Šmilauer et al. 2015). A value of 0.05 was chosen according to the results of
a numerical sensitivity analysis (Fig. S4).
**2.2.4 Input parameters**
In view of the preceding paragraph, the density of the ice grains was set to $\rho = f$ x 917 kg m$^{-3}$. The contact law parameters were
derived from typical values measured on ice. The Poisson coefficient $P$ was set to 0.3 (Schulson and Duval, 2009). The typical
Young's modulus $E$, the cohesion strength $C$ and the friction coefficient $tan(\varphi)$ values for the ice are usually evaluated around
1 x 10$^{10}$ Pa, 1 x 10$^6$ Pa and 0.2, respectively (Gammon et al., 1983; Schulson and Duval, 2009). For this study, a sensitivity
analysis to the values of these parameters was performed to get insights into their influence and best adjust simulation results
to the experimental measurements. The considered ranges were 1 x 10$^8$-1 x 10$^{10}$ Pa for $E$, 5 x 10$^5$-5 x 10$^6$ Pa for $C$ and 0.2-0.5
for $tan(\varphi)$, respectively. Note that the range of the Young's modulus $E$ ensures small grain overlaps, i.e. compliance with the
rigid grain assumption (Fig. S5). We must mention that, due to longer computing times, fewer parameter values could be
explored for large Young's modulus values. For the PP sample, no numerical simulations could be performed for a Young's
modulus of 1 x 10$^{10}$ Pa, as computing times were unreasonable ($E$ = 1 x 10$^8$ Pa, $t \sim$ 4 months and $E$ = 1 x 10$^9$ Pa, $t \sim$ 10 months
on a 72 cores machine with 2.6 GHz Intel Xeon processors (2.6 GHz) and 500 GB RAM. YADE scripts enable parallelisation
on up to 5 cores).

| Simulation setup | | |
|---|---|---|
| Sample width | $W$ | 13 mm |
| Sample height | $H$ | 15 mm |
| Tip radius | $R$ | 2.5 mm |
| Cone apex | $a$ | 60° |
| Tip velocity | $v$ | 20 mm s$^{-1}$ |
| Gravity | $g$ | 9.81 m s$^{-2}$ |
| **Numerical parameters** | | |
| Time step | $dt$ | $\sim$ 1 x 10$^{-6}$-1 x 10$^{-8}$ s |
| Mass factor | $f$ | 100 |
| Non-viscous damping coefficient | $\Lambda$ | 0.05 |
| **Material properties** | | |
| Grain density | $\rho$ | 917 x 10$^2$ kg m$^{-3}$ |
| Poisson coefficient | $P$ | 0.3 |
| Friction coefficient | $tan(\varphi)$ | 0.2–0.5 (default value 0.2) |
| Young's modulus | $E$ | 1 x 10$^8$–1 x 10$^{10}$ (default value 1 x 10$^9$) Pa |
| Cohesion | $C$ | 5 x 10$^5$–5 x 10$^6$ (default value 2 x 10$^6$) Pa |

**Table 2: Input parameters used for the simulations presented in this paper.**

## 2.3 Data processing

The outputs of the DEM simulations are the resisting force exerted by the grains on the penetrating rod and the displacement of the grains. These results can be directly compared to the experimental measurements.

### 2.3.1 Force sampling

The sum of the forces along the z-axis applied on all the facets constituting the penetrometer (cone and rod) is recorded at each time step. The characteristics of the raw numerical force profiles depend on the numerical parameters (notably the time step), and are not necessarily suited for direct comparison with experimental results. To obtain numerical profiles that can be compared to their experimental counterparts, the simulated force values were averaged over windows corresponding to displacement increments of 4 μm, thus matching the sampling frequency of the SMP. This averaging is also useful to smooth out high-frequency fluctuations linked to the very small time steps used in DEM. Finally, numerical and experimental force profiles are then re-sampled by linear interpolation over a regular grid with a step of 4 μm over the same depth. The profiles span from a depth of 0 mm (initial contact between the cone and the sample surface) to the chosen maximum depth, which, in our study, is set to 7 mm (i.e., 1750 points). This value corresponds to the minimum depth reached by the penetrometer during the experimental CPT tests for the selected samples.

### 2.3.2 Statistical indicators

Quantitatively, the DEM numerical model is evaluated by comparisons with experimental force profiles in terms of three statistical indicators: the mean macroscopic force $\bar{F}$ (N), the amplitude of force fluctuations $\sigma$ (N), and the correlation length $l$ (mm). The indicator $\sigma$ is calculated as the variance of the detrended force profile as follows:

$$\sigma = \bar{\tilde{F}}^2 , \qquad \tilde{F} = \frac{F - F_{sm}}{F_{sm}} \qquad\qquad (7)$$

with $\tilde{F}$ ([Eq. (5)], Peinke et al. 2019), the detrended force profile, $F$, the force profile and $F_{sm}$, the averaged force profile calculated over a rolling window $\Delta z = 3$ mm. The correlation length $l$ (mm) is also computed on the detrended force profile (Peinke et al. 2019). In our study, the snow samples exhibit a rather homogeneous structure allowing us to consider that $l$ is constant over the depth (Peinke et al., 2019). These three statistical indicators have been chosen because they are easily quantifiable and commonly used to describe force profiles obtained by CPT in snow (Johnsson and Schneebeli, 1999; Löwe and van Herwijnen, 2012; Peinke et al. 2019). In addition, they constitute key parameters to derive additional microstructural properties based on Poisson shot noise models (Löwe and van Herwijnen, 2012; Peinke et al. 2019).

To select the set of model mechanical parameters ($E$, $C$ and $tan(\varphi)$) providing the best fit to the experimental measurements, a total error $RE_{tot}$ is computed according to:

$$RE_{tot} = \sqrt{2\ RE_F{}^2 + RE_\sigma{}^2 + RE_l{}^2}$$ (8)
with $RE_k$ the logarithmic relative error calculated for the three statistical indicators, $k = (F, \sigma, l)$, as:
$$RE_k = \frac{log(measured\ value_k) - log(computed\ value_k)}{log(measured\ value_k)}$$ (9)
Given the difficulties in reproducing the correlation length with the DEM model for two out of four samples and the fact that
the values of the statistical indicators vary over several orders of magnitude (see Section 3.2), the logarithmic relative errors
$RE_k$ were computed with the log of the considered values. We have attributed a weight factor of 2 to the logarithmic relative
error $RE_F$ related to the mean macroscopic force, to put more emphasis on the correct reproduction of this quantity. Hence, for
each snow sample, the set of mechanical parameters minimising the total error $RE_{tot}$ was determined.
**2.3.3 Grain displacement analysis**
The position of all grains was recorded every ~0.4 mm of penetration in the DEM simulations. The total displacements and
the trajectories can therefore be reconstructed for each grain. Due to the thermodynamically active nature of snow, interrupted
experimental tests were not feasible and only the initial (before CPT) and the final states (after CPT) of the snow sample could
be imaged by μCT. Grain tracking, applied to the micro-tomographic images, has been performed by Peinke et al. (2020),
providing the total displacement of the identified grains. We thus compared the total displacement between the CPT
experiments and the DEM simulations at the same penetration depth, i.e., at the maximal penetration measured
experimentally. Note that grain tracking could not be performed for the PP sample due to the small size of the grains.
The profiles of vertical and radial displacements were averaged around the cone axis and over the height of an area located
between the top section of the cone and the sample surface. A displacement threshold of 0.03 mm was set to define the CZ
(Peinke et al., 2020). Only the radial profiles were compared to the experimental results, as we suspect the vertical profiles
derived from μCT scans might be misleading (Peinke et al. 2020). Indeed, before acquiring the post-CPT μCT scans, the tip
was removed from the snow. This procedure was performed about one hour after the tip penetration, to allow for bonds between
ice grains to re-form by sintering and limit grain displacements during tip removal. However, despite this precaution, some
grains in contact with the tip might have been dragged upward due to friction with the tip. Therefore,  the upward component
of the vertical displacement might have been overestimated in the experimental results, especially for the larger grains.
**3 Results**
**3.1 Simulated Cone Penetration Tests**
This section presents an example of CPT simulation results for the case of the RG snow sample with the following mechanical
parameters: $E = 1 \times 10^9$ Pa, $C = 5 \times 10^6$ Pa and $tan(\varphi) = 0.2$ (Table 3). The results for the other snow samples are shown in

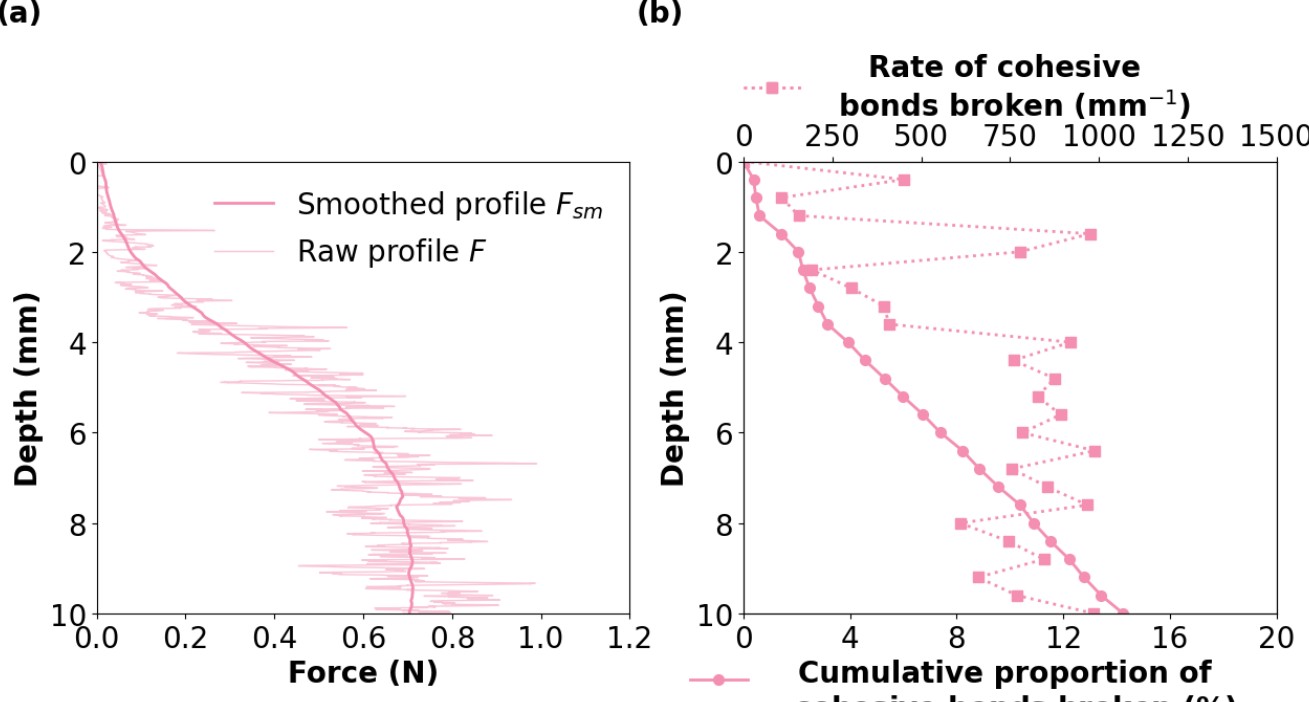

**Figure 2: (a) Force $F$ as a function of penetration depth (light line) obtained for the RG sample. The superposed smoothed profile**
**(bold line) $F_{sm}$ corresponds to the force value averaged over a rolling window of 3 mm. (b) Rate of cohesive bonds broken per unit**
**penetration depth and cumulative proportion of cohesive bonds broken (%) as a function of tip penetration depth. The initial**
**number of cohesive bonds is indicated in Table 1. The results are obtained with the mechanical parameters indicated in Table 3.**

The simulated penetration force globally increases with depth and is characterised by high-frequency fluctuations whose
amplitude also tends to increase with depth (Fig. 2 (a)). The force profile displays an 'S' shape with three stages: 1) up to ~
3.5 mm depth, the profile is convex, 2) between ~ 3.5 and ~ 6 mm depth, the increase of force with depth is almost linear, and
3) for depths larger than 6 mm, the force reaches a nearly constant value. A similar behaviour is observable for the RGlr and
PP samples (Fig. S6 (a) and S10 (a)), with slight variations in the transition depths between the different stages. For the DH
sample, the macroscopic force profile also displays stages 1 and 2, but the stabilisation at a nearly constant value is less evident
for the results presented in Fig. S (a). Stage 3 might be reached at greater depths for this sample.
The penetration of the tip induces bond failures in the simulated samples (Fig. 2 (b)). Overall, for the RG sample, about 15%
of the cohesive interactions broke over 10 mm of penetration, corresponding to an average rate of ~650 bond failures mm$^{-1}$.
This average bond failure rate is variable among the samples, reaching up to 1400 bond failures mm$^{-1}$ for RGlr sample (Figs.
S6 (b), S8 (b), S10 (b)). In detail, for the RG sample, we notice an increase in the bond failure rate at around 3.5 mm of
penetration depth (Fig. 2 (b)), coinciding with the transition between the first and second stages observed in the force signal
(Fig. 2 (a)). Bond failure intensity then remains nearly constant as the macroscopic force reaches its steady-state value. Similar
characteristics are observed for the other snow types (Figs. S6, S10) except for the DH sample, for which the slope change
between the first and second stages is less clear (Fig. S8 (b)).

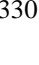


**Figure 3: (a) Simulated grain displacement map for the RG sample. The red arrows indicate the grain trajectories while the tip is**
**penetrating (sampling = 0.4 mm). White grains correspond to grains that are not represented in the DEM simulation. The final tip**
**position is indicated by the black solid lines. The horizontal black dashed line indicates the cone top. (b) Radial (upper panel) and**
**vertical (lower panel) displacement profiles (red curves) for the RG sample. These profiles represent averages computed from the**
**sample surface to the cone top. By convention, downward (respectively upward) movement corresponds to positive (respectively**
**negative) values of vertical displacement. The shadowed areas around the solid lines represent the standard deviation of grain**
**displacements. The results are obtained with the mechanical parameters indicated in Table 3.**
Figure 3 (a) shows the total displacement of the grains as well as grain trajectories. The largest displacements (up to several
mm) are observed for grains initially located on the path of the tip. Around the tip, the displacements are < 1 mm and are
mainly localised close to the tip. Grain trajectories indicate that grains are pushed downward from each side of the tip. Grains
initially located on the tip axis display quasi-straight vertical trajectories. The trajectories become more radial and curved away
from the tip medial axis, with grains also being pushed aside. Both radial and vertical displacement profiles show a pronounced
decreasing trend and reach almost zero values at a radial position of about 1.7-1.8$R$ (Fig. 3 (b)). The vertical profile attests of
a dominant downward movement of the grains close to the tip. Similar observations are made for the DH (Fig. S9) and PP
(Fig. S11) samples. In contrast, for the RGlr sample, vertical displacements are smaller and oriented slightly upward on
average, for the mechanical parameters chosen here (Fig. S7).
**3.2 Sensitivity to mechanical parameters**
The influence of the mechanical parameters (Young's modulus, cohesion, friction coefficient) involved in the contact law has
been systematically explored. For the RG sample, the force profiles obtained for the different values of the parameters within
the explored ranges (Table 2) are presented in Figure 4, and synthetic plots of the sensitivity of the statistical indicators to
these parameters are presented in Figure 5. The results for the other snow samples can be found in Sect. S2.3. Table S3 also
summarises the values of statistical indicators in all cases.

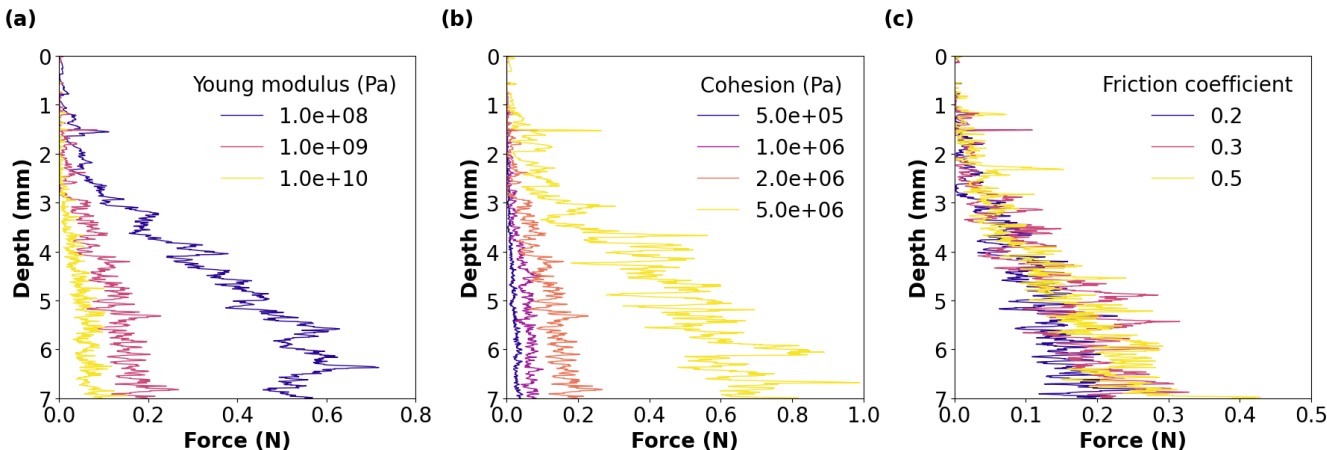

**Figure 4: Influence of mechanical parameters on the simulated force profile. The sensitivity analysis has been performed on (a)**
**Young's modulus $E$ (Pa) (for $C = 2.0 \times 10^6$ Pa and $tan(\varphi) = 0.2$), (b) cohesion $C$ (Pa) (for $E = 1.0 \times 10^9$ Pa and $tan(\varphi) = 0.2$), and (c)**
**friction coefficient $tan(\varphi)$ (for $E = 1.0 \times 10^9$ Pa and $C = 2.0 \times 10^6$ Pa). The results presented here correspond to the RG sample.**

First, it can be observed that increasing Young's modulus decreases the mean macroscopic force (Figs. 4 (a) and 5 (a)) and the
correlation length (Fig. 5 (c)). The influence of Young's modulus on the amplitude of force fluctuations is more complex and
displays a co-dependency with the cohesion values (Fig. 5 (b)). For low (respectively high) cohesion values, the amplitude of

force fluctuations shows a decreasing (respectively increasing) trend with Young's modulus. Regarding the influence of cohesion, it is observed that increasing this parameter increases the three statistical indicators. Finally, increasing the friction coefficient, generally also leads to an increase of the three statistical indicators. Note however that, over the range of explored friction coefficient values (0.2-0.5), the sensitivity to this parameter is less important than for the other two mechanical parameters (where $E$ is varied over two orders of magnitude and $C$ is varied over one order of magnitude). Despite changes in absolute force values, the evolution of the force profiles (Figs. S14, S18 and S22) and statistical indicators (Figs. S15, S19 and S23) with the mechanical parameters follow similar trends for all the samples.

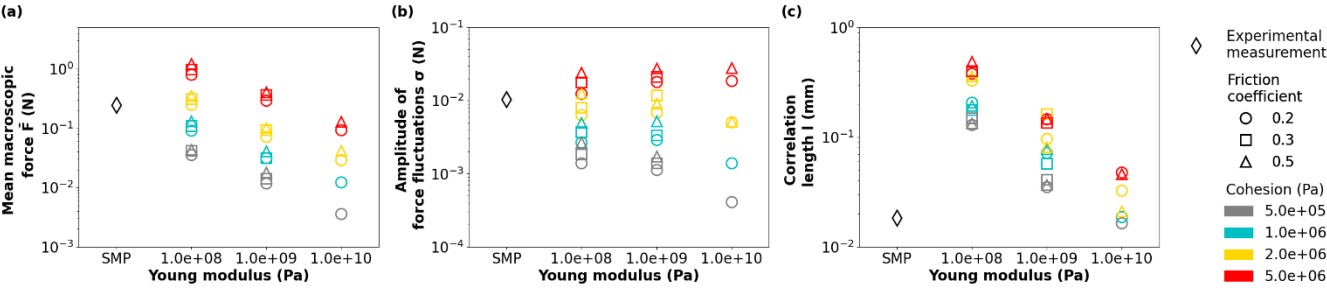

**Figure 5: Evolution of statistical indicators as functions of Young's modulus, cohesion and friction coefficient: (a) Mean macroscopic force $\bar{F}$, (b) amplitude of force fluctuations $\sigma$, and (c) correlation length $l$. The experimental results (black diamonds) are also represented in the plots. The results presented here correspond to the RG sample.**

The number of broken bonds per increment of tip penetration depth appears rather insensitive to Young's modulus (Figs. S12 (a), S16 (a), S20 (a), S24 (a)) and is only slightly reduced when cohesion increases (Figs. S12 (b), S16 (b), S20 (b), S24 (b)). Conversely, this quantity is significantly affected by the friction coefficient, with an increase of the average bond failure rate when $tan(\varphi)$ increases (Figs. S12 (c), S16 (c), S20 (c), S24 (c)).

Finally, it is observed that the influence of the mechanical parameters on the radial grain displacement profiles is negligible (Figs. S13, S17, S21, S25). Young's modulus shows no influence on the vertical grain displacement either. Cohesion appears to play a role in the vertical displacement profile for the RGlr sample, by enhancing upward movements. Larger friction coefficients tend to increase the downward movement of the grains close to the tip for all the snow types.

**3.3 Comparison of DEM results with experimental measurements**

A first noticeable observation is that, for the values of the mechanical parameters tested, the orders of magnitude of the statistical indicators obtained numerically are consistent with the experimental results in most of the cases (Figs. 5, S15, S19, S23, Table S2, Table S3). This demonstrates that the DEM model is indeed capable of reproducing the main characteristics of the CPT force profile (Fig. S26, Table S2). However, we highlight the difficulty of matching the three statistical indicators at once for a given combination of the three mechanical parameters studied. Hence, for the RG sample (Fig. 5), the DEM

simulation can reproduce the experimental mean macroscopic force and the amplitude of force fluctuations but tends to overestimate the correlation length by a factor of 8 for the best combination of mechanical parameters. For the RGlr and DH samples (Figs. S15, S18), all the experimental statistical indicators can be reproduced individually, but not for one single combination of the mechanical parameters. For the PP sample, the experimental mean macroscopic force and the amplitude of force fluctuations can be reproduced numerically, but the correlation length is systematically overestimated by a factor of at least 8 (Fig. S23).

| Sample | E (Pa) | C (Pa) | tan(φ) | $RE_F$ | $RE_\sigma$ | $RE_l$ | $RE_{tot}$ |
|--------|--------|--------|--------|--------|-------------|--------|------------|
| RG | $1 \times 10^9$ | $5 \times 10^6$ | 0.2 | $1.2 \times 10^{-1}$ | $1.2 \times 10^{-1}$ | $5.2 \times 10^{-1}$ | $5.6 \times 10^{-1}$ |
| RGlr | $1 \times 10^9$ | $1 \times 10^6$ | 0.3 | $5.5 \times 10^{-2}$ | $-4.6 \times 10^{-1}$ | $1.1 \times 10^{-1}$ | $4.8 \times 10^{-1}$ |
| DH | $1 \times 10^{10}$ | $5 \times 10^6$ | 0.2 | $1.2 \times 10^{-1}$ | $-1.1 \times 10^{-1}$ | $-2.3 \times 10^{-1}$ | $3.1 \times 10^{-1}$ |
| PP | $1 \times 10^9$ | $2 \times 10^6$ | 0.5 | $-1.3 \times 10^{-1}$ | $-1.6 \times 10^{-1}$ | $6.5 \times 10^{-1}$ | $6.9 \times 10^{-1}$ |

**Table 3: Selected combination of mechanical parameters for RG, RGlr, DH and PP samples. The indicated values of Young's modulus $E$, cohesion $C$ and friction coefficient $tan(\varphi)$ correspond to the combinations that yield the lowest total error $RE_{tot}$ on the statistical indicators (mean macroscopic force $\bar{F}$, amplitude of force fluctuations $\sigma$, correlation length $l$) measured experimentally. Logarithmic relative error $RE_k$ for all the mechanical parameter combinations tested are indicated in Table S3.**

Based on the sensitivity analysis (Sect. 3.2.3.), we selected for each sample the combination of the three mechanical parameters that minimises the total error $RE_{tot}$ (Tables 3, S3). The corresponding simulated force profiles (referred to as 'Numerical simulation 1') are compared with the experimental profiles in Fig. 6. Note that the error values quoted in the text below correspond to relative errors calculated without the logarithmic function, as they are easier to grasp. These values therefore differ from the logarithmic relative errors shown in Tables 3 and S3 and used for the parameter selection. From a qualitative point of view, a good overall agreement is observed between these numerical and experimental force profiles. For the RG sample, the experimental mean macroscopic force is overestimated by ~20% by the numerical result, the amplitude of force fluctuation is overestimated by ~70% and the correlation length is largely overestimated by a factor of 8 (Figs. 5, 6 (a), Table 3). Both the experimental and numerical force profiles reach a quasi-steady-state value at about the same depth (~6 mm, S27). For the RGlr sample, the experimental mean macroscopic force is fairly reproduced with a relative error of 6%, the amplitude of force fluctuations is underestimated by ~60% and the correlation length is overestimated by 35% (Figs. S15 and 6 (b), Table 3). We note that the slope change between 2.5 and 3 mm penetration depth is reproduced numerically. However, it appeared difficult to reproduce numerically the amplitude of force fluctuations in the upper section (from 0 to 4 mm) of the experimental profile. For the DH sample, the experimental mean macroscopic force is overestimated by ~25%. The experimental amplitude of force fluctuations is underestimated by 28% and the correlation length is about half of the experimental value (Figs. S19, 6 (c), Table 3). The numerical results minimise the force peaks observed in the upper part of the experimental profile (above 3 mm) but reproduce fairly well the main features of the amplitude of force fluctuations, especially the force "jump" at 3 mm depth. Finally, for the PP sample, the experimental mean macroscopic force is underestimated by ~30%, while the experimental amplitude of force fluctuations is underestimated by ~60%. In this case, the

experimental correlation length could not be reproduced at all, with values overestimated by a factor of 20 (Figs. S23 and 6
(d), Table 3).

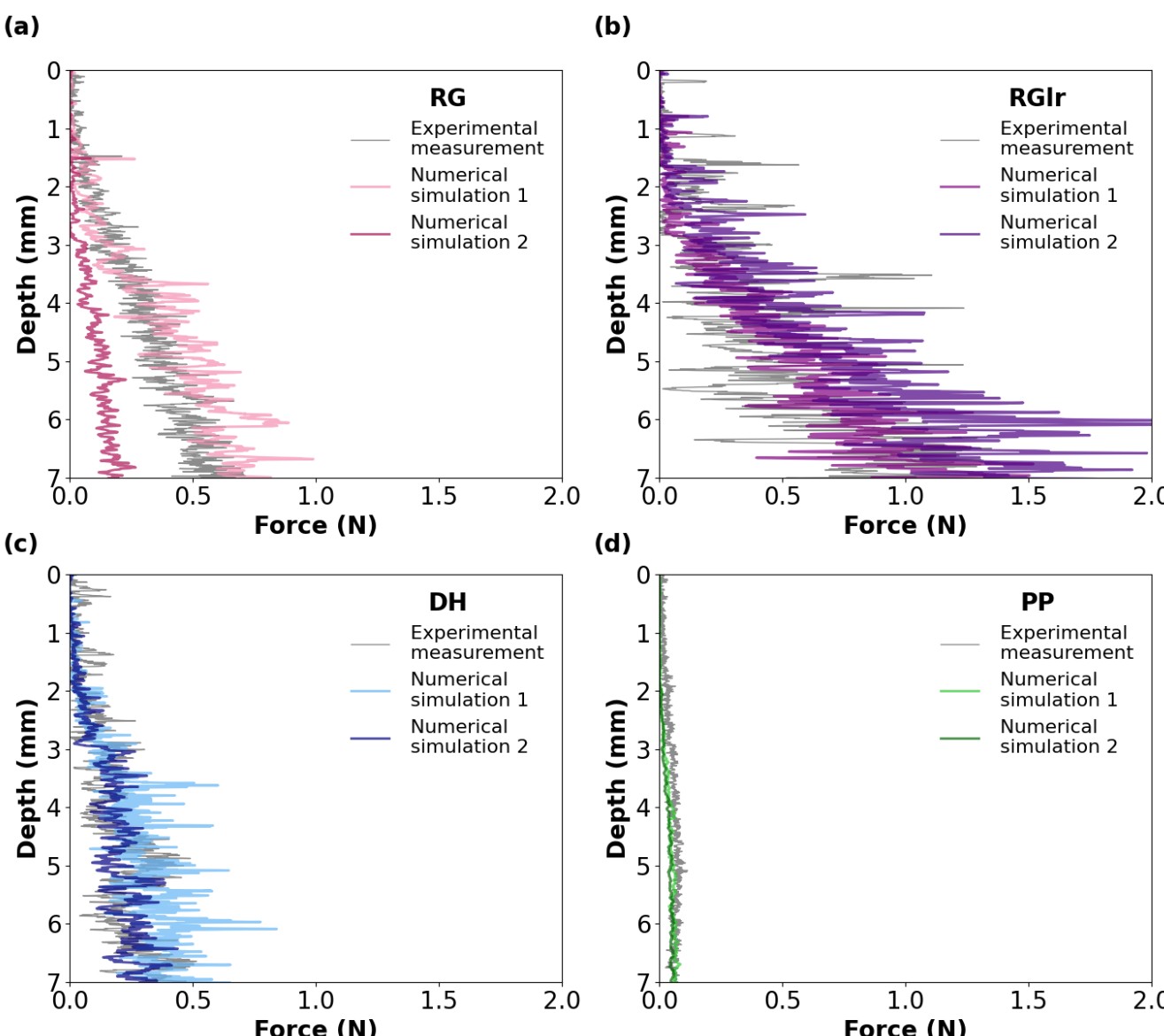


Figure 6: Experimental (grey) and numerical (coloured) CPT force profiles obtained for (a) RG, (b) RGlr, (c) DH, and (d) and PP
samples. The "Numerical simulation 1" profiles correspond to the best fit of the mechanical parameters determined for each sample
(Table 3), while "Numerical simulation 2" profiles correspond to an overall best fit of the mechanical parameters for the four
samples ($E$ = 1 x 10$^9$ Pa, $C$ = 2 x 10$^6$ Pa and $tan(\varphi)$ = 0.2, Table S3).

For comparison, we also selected the single set of mechanical parameters that minimises the combined total error $RE_{tot}$ on RG,
RGlr, DH and PP samples. Corresponding values are: $E = 1 \times 10^9$ Pa, $C = 2 \times 10^6$ Pa and $tan(\varphi) = 0.2$. The respective
logarithmic relative errors for each sample can be found in Table S3. As before, the error values presented in the text below
correspond to the relative errors without the logarithmic function applied to the values. In general, the corresponding simulated
force profiles (referred to as 'Numerical simulation 2' in Fig. 6) also show a fair agreement with the experimental results. For
the RG sample, however, the experimental mean macroscopic force is significantly underestimated by ~70% (Figs. 5, 6 (a),
Table S3). The numerical amplitude of force fluctuations is underestimated by ~35%, while the correlation length is
significantly overestimated by a factor of 5. For the RGlr sample, the agreement is acceptable for the three statistical indicators
with relative errors around 50%. For the DH sample, the experimental mean macroscopic force is reproduced at 90%, while
the experimental amplitude of force fluctuations is underestimated by 60% and the experimental correlation length is
overestimated by a factor of ~2. Finally, for the PP sample, the experimental mean macroscopic force is underestimated by
~80%, the amplitude of force fluctuations by ~85% and the experimental correlation length is again strongly overestimated by
a factor of 20 (Figs. S23 and 6 (d), Table S3).

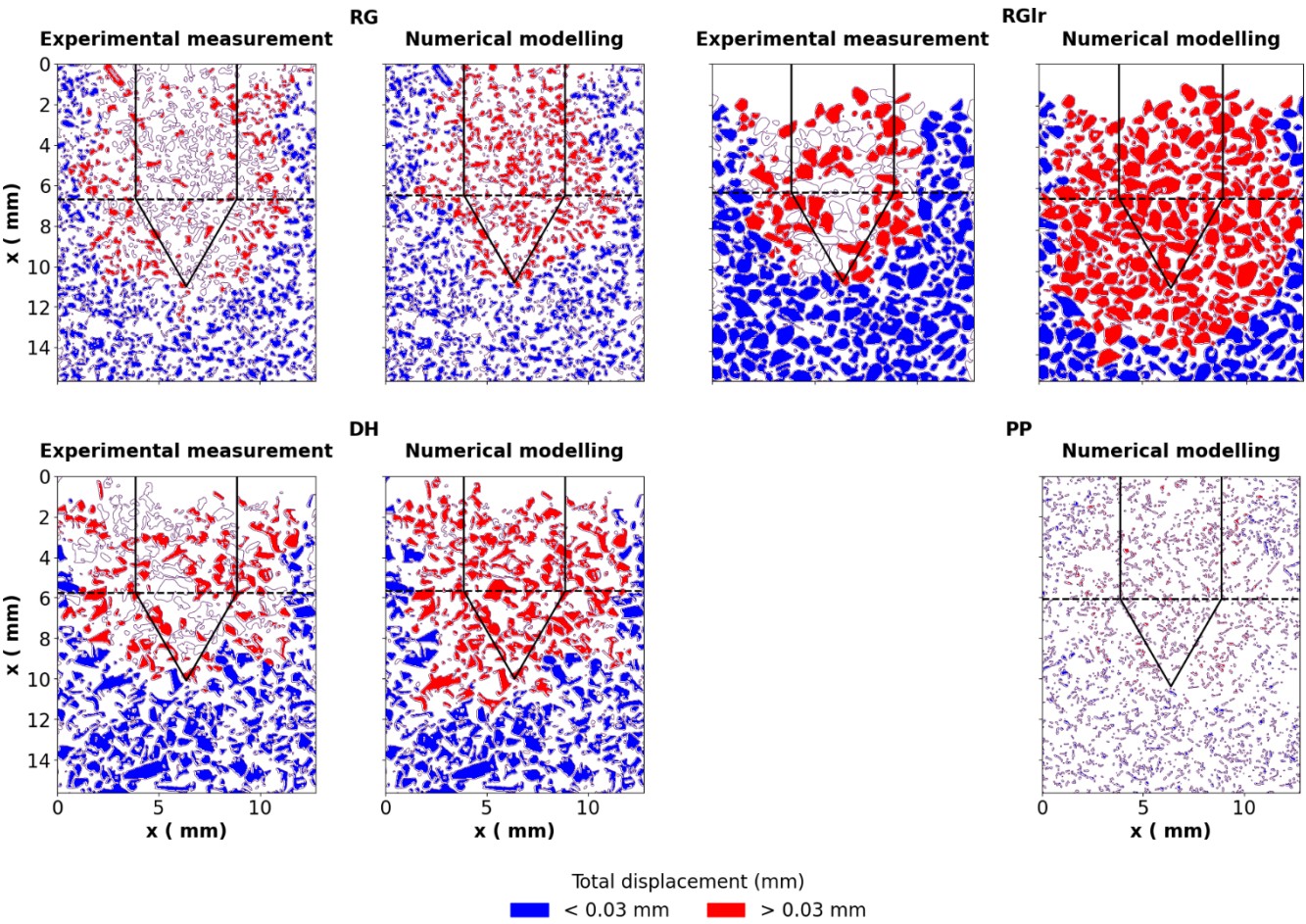

Total displacement (mm)
🟦 < 0.03 mm  🟥 > 0.03 mm


**Figure 7: Total displacement maps obtained experimentally with μCT (left panels) and numerically with DEM simulation (right panels) for the RG, RGlr, DH and PP samples. A displacement threshold of 0.03 mm has been set to define the deformation zone (Peinke et al., 2020). White grains correspond to non-trackable grains in μCT scans (Peinke et al., 2020) and grains not represented in the DEM simulations. The final tip position is indicated with black solid lines. The horizontal black dashed line indicates the cone top. Displacement profiles shown in Fig. 8 are computed from the sample surface to the cone top. Numerical results are obtained with the mechanical parameters indicated in Table 3. The experimental displacement field could not be determined for the PP sample.**

453

As shown in Fig. 7, the DEM simulations also proved capable of reproducing, at least qualitatively, the experimental grain displacement patterns derived from μCT scans for the four snow types. Essentially similar results are obtained with the individual best-matching sets of mechanical parameters indicated in Table 3 (Fig. 7), and with the globally-matching set of parameters introduced in the previous paragraph (Fig. S28). For the RG sample, the overall shape and size of the deformation zone are well reproduced by the simulations. For the DH sample, the radial extension of the deformation zone is well reproduced by the simulations, but the vertical extension tends to be overestimated. The largest discrepancies are observed for

the RGlr sample, for which the radial and vertical extensions of the deformation zone are overestimated compared to the experimental data.

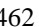

**Figure 8: Radial displacement profiles (solid lines) obtained experimentally (black) and numerically (coloured) for the RG, RGlr, DH and PP samples. The shadowed areas around the solid lines correspond to the standard deviation of grain displacement and exhibit the variability of the radial displacement of grains. The numerical results are obtained with the mechanical parameters indicated in Table 3.**

Similarly, the radial displacement profiles obtained from the DEM numerical simulations are overall in good agreement with their experimental counterparts (Figs. 8 and S29). Consistently with the displacement maps, the largest discrepancy is observed

for the RGlr sample. In particular, the abrupt slope break seen in the experimental profile at a radial position of about 1.5 is
not reproduced in the numerical profile. Note however that, due to a relatively low number of trackable grains (Fig. 7), the
standard deviation of the grain radial displacements is larger in the experimental measurements, which may result in a larger
uncertainty on the average profile. In contrast, simulations on the RG and DH samples show a very good agreement with the
experiments. The CZ (defined with displacement threshold set at 0.03 mm) obtained from numerical simulations extends
radially up to 1.6$R$, 2.2$R$, 2.0$R$ and 1.5$R$ for the RG, RGlr, DH and PP samples, respectively. In comparison, the CZ derived
from μCT scans extends radially up to 1.7$R$, 1.5$R$ and 1.9$R$ for the RG, RGlr and DH samples, respectively (no measurement
for PP sample).
**4 Discussion**
**4.1 Evaluation of the DEM model**
We used three mechanical parameters, namely Young's modulus, the cohesion and the friction coefficient, to adjust the
simulated force profiles to the experimental results. Overall, the numerical model could reproduce relatively well the
mechanical response of all studied numerical samples with a single set of mechanical parameters ($E = 1$ x $10^9$ Pa, $C = 2$ x $10^6$
Pa and $tan(\varphi) = 0.2$) (Fig. 6), indicating that the differences in the force profiles among the samples are mainly dependent of
the snow microstructure.
It should also be noted that the values of the mechanical parameters obtained by adjusting the model on the experimental data
(either globally for all samples or for each sample individually, Table 3) are reasonably close to the mechanical properties of
ice. Young's modulus of ice is measured between 9 x $10^9$ Pa and 10 x $10^9$ Pa (Gammon et al., 1983), while our selected values
range between 1 x $10^9$ Pa and 1 x $10^{10}$ Pa. Recall that, in YADE, the Young's modulus is a numerical parameter used to define
the normal contact stiffness, and is not expected to necessarily correspond to the physical Young's modulus of the material
(Sect. 2.2.2). Nevertheless, the fact that the numerical value of $E$ is in the same range of magnitude as the elastic properties of
ice provides confidence that the DEM model and the used contact law ([Eqs. (1)-(5)]) correctly capture the physical processes
at play. Similarly, the numerical cohesion values, ranging between 1 x $10^6$ Pa and 5 x $10^6$ Pa, are in agreement with typical
cohesion values measured on ice (in the range 2 x $10^6$ Pa to 6 x $10^6$ Pa, Schulson and Duval, 2009). Finally, numerical friction
coefficients appear to be on the order of 0.2–0.5, while values measured experimentally generally range from 0.02 to 1 (Fish
and Zaretsky, 1997; Maneno and Arakawa, 2004). All these results reinforce the confidence in the relevance of the DEM
model.
We acknowledge that the mechanical parameters obtained from minimising the logarithmic relative errors on the statistical
indicators do not necessarily represent optimal values, in the sense that only a limited number of parameter sets could be tested.
Based on the sensitivity analysis, a more proper inversion procedure could be developed to retrieve true optimal values of the
mechanical parameters. This would certainly provide more robust elements as to whether a single set of mechanical parameters
can be used to represent the experimental results of all snow types, or whether these mechanical parameters differ according
to the snow type. Our current analysis cannot provide a conclusive answer to this question. Note that ice is a polycrystalline
material, whose mechanical behaviour can be strongly anisotropic depending on the ice structure (Fish and Zaretsky 1997;
Thorsteisson, 2001; e.g. Maeno and Arakawa, 2004). Therefore, it is not unlikely that ice bonds between grains could be
characterised by different mechanical properties depending on the specific conditions of snow formation and evolution.
As further proof of DEM predictive capabilities, we could also observe that the grain displacement fields measured for the
different snow types were overall well reproduced by the simulations (Figs. 7 and 8). In particular, the model captures the
radial extent of the deformation zone, which is on the order of 1.5$R$-2.2$R$. A discrepancy between the numerical and
experimental radial displacement profiles was observed for the RGlr sample. However, it can be noted that these experimental
radial displacement profiles for the RGlr sample also show the largest divergence with the prediction of the cavity expansion
model (CEM) (Yu and Carter, 2002), as shown by Peinke et al. (2020). In fact, the radial profile predicted by the CEM for this
sample is similar to the radial profile obtained numerically in this study.
**4.2 Interpretation**
**4.2.1 Sensitivity to the mechanical parameters**
The sensitivity analysis revealed a strong influence of the mechanical parameters on the simulation results. In particular, a
clear dependence of the mean macroscopic force with Young's modulus $E$ was observed, suggesting that a significant part of
the sample undergoes elastic deformation, while brittle failures are confined in a region close to the tip. Note that a similar
dependence to $E$ with a cohesive contact law has been observed in DEM modelling of soil compression (De Pue et al., 2019)
and snow compression (Bobillier et al., 2020). The mean macroscopic force, the amplitude of force fluctuations and the
correlation length all increase with the cohesion $C$ and, to a smaller extent, with the friction coefficient $tan(\varphi)$. This can be
related to the fact that increasing cohesion and friction between grains increase bond strength. It was also observed that
cohesion tends to prevent bond failures and to favour the upward movement of grains for samples with a large initial density,
such as RGlr. In contrast, increasing the friction coefficient enhances the bond failure rate and the downward movement of
grains (Figs. S12, S16, S20, S24). When sliding between grains is inhibited, a grain dragged by the tip movement will entrain
surrounding grains more easily, thus enlarging the deformation zone and triggering additional bond failures. Finally, radial
grain displacements and the radius of the deformation zone appeared to be mostly insensitive to the mechanical parameters,
indicating that these features are mainly controlled by CPT configuration and snow microstructure.
**4.2.2 Compaction zone development**
For all snow types, the force profiles computed numerically display a 'S' shape (Figs. 1, S6, S8, S10). We attribute this shape
to the development of a compaction zone (CZ) in front of the tip during its penetration into the numerical sample. More
specifically, the first stage of the force profiles (slope increase) is presumably caused by the progressive entry of the cone into
the sample. The second stage (constant slope) is attributed to the development of the CZ in front of the tip. The third stage
(quasi-constant force value) suggests that a steady-state regime, with a fully-developed CZ, is reached. Depending on the snow
type, the numerical results indicate that full development of the CZ occurs for 6 mm to 8 mm of penetration depth. These
results agree with the experimental profiles for the RG, DH and PP samples. Globally, we can highlight that the DEM
simulations are able to reproduce fairly well the global shape of the experimental profiles, and thus to correctly capture the
development of the CZ.
Nevertheless, in another experimental study, the CZ has been reported to be fully developed only for around 40 mm of depth
penetration (Herwijnen, 2013), which is significantly deeper than the experimental and numerical results obtained in this study.
A first hypothesis to explain this discrepancy is that since the maximum depth of our CPT force profiles is 10 mm, we might
miss information on the full CZ development. A second explanation could be related to the differences in the experimental
setups. Indeed, Peinke et al. (2020) performed CPT on snow samples contained in cylinders of 20 mm diameter and 20 mm
height, which is significantly smaller than the decimetric snow samples considered by Herwijnen (2013). Boundary effects
might thus play a role in limiting the development of the CZ. Finally, the tip geometry also differs between the two studies.
Peinke et al. (2020) used a plain tip, while Herwijnen (2013) used the original SMP tip geometry with a cone radius larger
than the rod. A sensitivity analysis comparing the two geometries showed an influence over the upper 12 mm of the force
profiles (Peinke, 2020). The plain tip geometry resulted in larger values of the mean macroscopic force and the amplitude of
force fluctuations values. This effect might also influence the characteristics of the CZ development, which could be studied
in the future using the presented numerical model.

### 4.2.3 Grain-tip interaction

The sensitivity analysis to the grain shape representation (Sect. S1.1) provides interesting insights into the interpretation of
force profiles. In particular, the study highlighted that the grain shape representation could be relatively coarse (high volumetric
error $E_V$) but still produce a force profile with an acceptable mechanical error $E_M$ compared to a reference profile obtained for
a fine grain shape representation ($E_V < 10\%$) (Fig. S1, Table S1). This is notably the case for the RG sample, for which the
selected grain shape representation ($L = 5$, $S = 0.3$) corresponds to a value of $E_V$ of about 40%. Large values of $E_V$ often imply
grain loss, as the smallest grains identified in the μCT scans cannot be represented by the DEM with coarse spherical elements.
Yet, the similarity of the force profile to the reference force profile indicates the limited contribution of these smallest grains
to the macroscopic force, compared to the largest grains with stronger bonds. The loss of grains and bonds might nevertheless
directly affect the force fluctuations, providing a potential explanation as to why the DEM model underestimates the correlation
length obtained experimentally for the samples with the smallest grain sizes (RG and PP) (Figs. 5, S23).

### 4.2.3 Scaling relation for the mean macroscopic force

To try and synthesise the large number of simulation results obtained in this study, scaling relations describing the evolution
of the statistical indicators as a function of the main simulation parameters can be looked for. We focused in particular on the
mean macroscopic force $\bar{F}$, which was observed to depend both on the mechanical parameters $E$, $C$ and $tan(\varphi)$, as well as on

sample microstructure. Since the range of friction coefficient values (between 0.2-0.5) that we could explore remained limited compared to the ranges of $E$ and $C$, the parameter $tan(\varphi)$ was not included in this analysis and the results presented below correspond to a single value $tan(\varphi) = 0.3$.

First, inspection of our results (see Figs. 5 (a), S15 (a), S19 (a), S23 (a)) indicates that the dependencies of the mean macroscopic force $\bar{F}$ to the Young's modulus $E$ and cohesion $C$ appear to be consistent across the four tested samples (see also Table S4). More precisely, $\bar{F}$ scales with $E$ according to a power law of the form $\bar{F} \sim C^{-\alpha}$, with an exponent $\alpha$ on the order of 1/2. Similarly, $\bar{F}$ scales with $C$ according to a power law of the form $\bar{F} \sim C^{\beta}$, with $\beta$ on the order of 3/2.

Second, we can expect $\bar{F}$ to be also related to the rate of cohesive broken bonds per unit penetration depth. In particular, it is observed (see Figs. S12, S16, S20, S24) that the slope $\lambda$ of the cumulative proportion of broken bonds as a function of depth is essentially independent of the Young's modulus and cohesion. Conversely, as shown in Fig. 9 (a), this slope $\lambda$ is linearly related to the initial contact density $\nu$ defined as:

$$\nu = z\Phi \tag{10}$$

with $z$ the coordination number (number of initial cohesive interactions between grains divided by the number of grains, see Table 1) and $\Phi$ the volume fraction of the sample (ice density = 917 kg m$^{-3}$, see Table 1).

From these different observations, the following scaling law for the mean macroscopic force $\bar{F}$ can be proposed:

$$\bar{F} = B\,T\,C\left(\frac{C}{E}\right)^{\alpha} f(\nu) \tag{11}$$

with $B$ a dimensionless constant, $T$ (m²) the surface area of the cone (with a radius $R$ and a cone apex $a$, Table 2) in contact with the sample, and $f$ a function to be determined. Figure 9 (b) shows the dimensionless quantity $\bar{F}T^{-1}E^{1/2}C^{-3/2}$ plotted against the initial contact density $\nu$. We observe that all the simulation results for the four snow types and the different values of Young's modulus and cohesion nicely merge on a unique logarithmic trend. Note, however, that a relatively larger dispersion is observed for RGlr ($\nu = 1.63$) compared to the other samples.

Equation (11) encapsulates in a single relation the main physics controlling the mean macroscopic force recorded by the penetrometer. In particular, this relation indicates that the influence of snow microstructure can be captured, at least as a first approximation, by the initial contact density $\nu$. Former studies already showed that this parameter plays a key role in the mechanical behaviour of cohesive granular materials (Gaume et al. 2017). Looking for similar relations describing the other statistical indicators (amplitude of force fluctuations and correlation length) constitutes an interesting prospect for future analyses, although we can anticipate these indicators to display more complex dependencies. Further analyses will also be required to explore the influence of the friction coefficient on these relations.

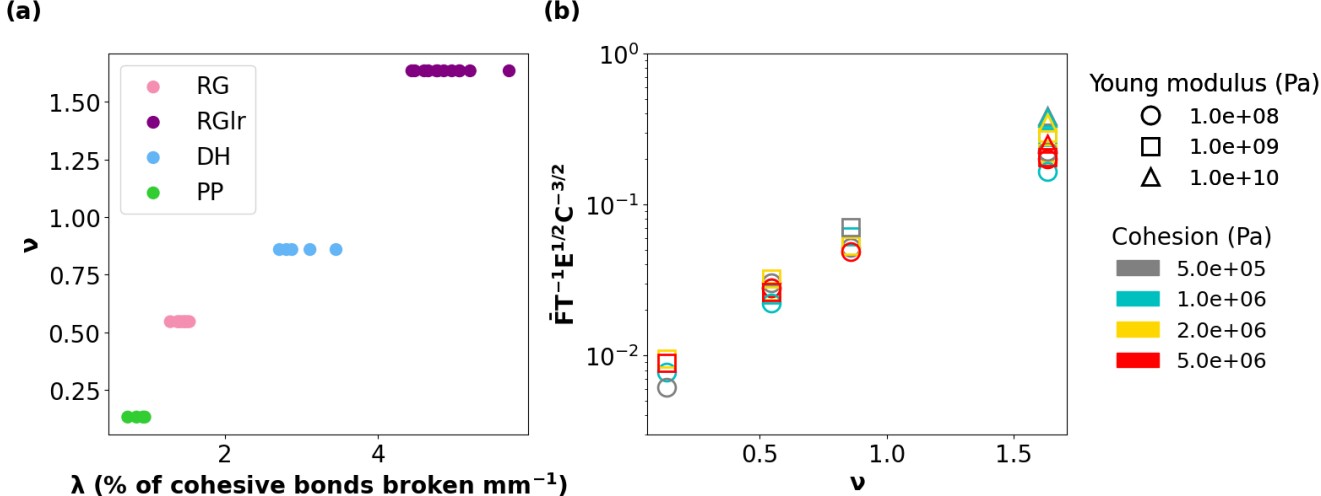

**(a)**

ν

1.50
1.25
1.00
0.75
0.50
0.25

RG
RGlr
DH
PP

λ (% of cohesive bonds broken mm⁻¹)

2    4

**(b)**

$\bar{F}T^{-1}E^{1/2}C^{-3/2}$

$10^0$
$10^{-1}$
$10^{-2}$

ν

0.5    1.0    1.5

Young modulus (Pa)
○  1.0e+08
□  1.0e+09
△  1.0e+10

Cohesion (Pa)
5.0e+05
1.0e+06
2.0e+06
5.0e+06

**Figure 9: (a) Initial contact density $\nu$ versus the slope $\lambda$ of the proportion of cohesive bonds broken per unit depth (mm⁻¹) for each snow type. The values of initial contact density $\nu$ were computed with Eq. (10) and the values indicated in Table 1. The slopes $\lambda$ were computed from the evolution of the cumulative proportion of cohesive bonds broken (Figs. S12, S16, S20, S24) over a window of 7 mm depth. (b) Dimensionless quantity $\bar{F}T^{-1}E^{1/2}C^{-3/2}$ (see Eq. (11)) versus the initial contact density $\nu$ for all simulation results. All the results are provided for a friction coefficient $tan(\varphi)$ of 0.3.**

## 5 Conclusion

We have evaluated a numerical model based on DEM that reproduces the mechanical behaviour of snow in the brittle regime. The DEM model takes into account the ice properties and the snow microstructure captured by tomography. The experimental configuration of the CPT measurements conducted on different snow types by Peinke et al. (2020) has been reproduced with the DEM model. Three parameters namely, the mean macroscopic force, the amplitude of force fluctuations and the correlation length, were used to quantify the similitude of the numerical and experimental profiles. The grains displacement field was computed and compared to the experimental displacement field derived from μCT scans acquired before and after the CPT. The DEM model has demonstrated overall a good capability to reproduce the mechanical responses of CPT performed in different snow types. The computed force profiles satisfactorily reproduce the main characteristics of the experimental force profiles. The results revealed that the force profile characteristics are strongly dependent on the microstructure. A sensitivity analysis also demonstrated the dependence of the mechanical response to the mechanical parameters of the contact law. In particular, a simple scaling law could be derived relating the mean macroscopic force computed by the DEM to the mechanical parameters $E$ (Young's modulus) and $C$ (cohesion) and to the microstructure characteristics captured by the initial contact density. The displacement fields are also well reproduced by the model, except for the RGlr sample showing a larger extent for the numerical results. The agreement in terms of radial displacement profiles is very good. The grains are mainly travelling

downward during the CPT, although for the RGlr sample, the upward movements close to the surface are not negligible. The
CPT implies a complex deformation field with a compression zone around the apex and an expansion zone close to the surface
(Peinke et al., 2020). Therefore being able to reproduce the force profiles (including high-frequency fluctuations) and
displacement fields for this mechanical test constitutes a strong validation of the reliability of the DEM model.
However, a downside of the DEM method is its high computational cost (simulation times ranging between 1 week to several
months depending on the physical and numerical parameters for the chosen CPT configuration), which limited the range of
mechanical parameters that could be explored for all snow types. The developed DEM model nonetheless constitutes a versatile
approach that could be applied to various materials and configurations in future studies. In particular, it will be possible to use
the model to gain more physical insights into the interaction between the tip and the grains, in order to better interpret the CPT
force profiles. Such analyses will provide ways to test and derive relevant macro- and micromechanical parameters to
characterise the microstructure properties from the CPT force signal solely. In particular, the validity of the assumptions made
by the HPP-NHPP method, as well as the influence of the CZ development, will be assessed. Future studies may also consider
refining the used contact laws to investigate, e.g. the influence of sintering processes on CPT results.

## Code availability

Codes can be provided by the corresponding author upon request.

## Data availability

All data can be provided by the corresponding author upon request.

## Author contribution

CH, PH and GC developed the numerical model, CH performed simulations and evaluated the numerical model, IP, PH, GC,
JR designed experiment, IP acquired experimental data, IP processed and analysed experimentation measurements, CH
analysed and interpreted numerical results, CH wrote the manuscript draft, PH and GC reviewed and edited the manuscript.
**Competing interests**
GC is a member of the editorial board of The Cryosphere. The peer-review process was guided by an independent editor, and
the authors have no other competing interests to declare.
**Acknowledgements**
This work benefited from financial supports from the Centre National de la Recherche Scientifique (CNRS), the Centre
National de la Recherche Météorologique, the Agence Nationale de la Recherche (ANR project MiMESis-3D ANR-19-CE01-
0009). IGE and CNRM-CEN are parts of LabEx OSUG (ANR-10-LABX-0056). IGE is part of LabEx TEC21 (ANR-11-
LABX-0030). We thank the two reviewers, Richard Parson and Henning Löwe, for their constructive feedback that enabled
us to significantly improve the quality of our manuscript.

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
