# Peer review of "Microstructure-based modelling of snow mechanics: experimental evaluation on the cone penetration test"

_The Cryosphere, 2023_

## Referee Comment (RC2)

**Review of "Microstructure-based modelling of snow mechanics: experimental evaluation on the cone penetration test" by Herny et al**

**Main comments**

The present paper is a relevant contribution towards understanding cone pentration tests of snow from microstructure based modelling. It includes many novel aspects and warrants publication after a little polish.

Main points:

- What is the influence of averaging the simulation data on fixed $4\mu$m windows? To fully understand the simulations it would be helpful to include also the raw (non-smoothed) force signals alongside with the $4\mu$m-window versions that are finally used to evaluate the statistical descriptors and compared to the experiments. The main question here is: Can the authors confirm that the results reported in l345 (i.e. the "more complex" behavior of the force standard deviation and the correlation lengths (Fig 5b,5c and corresponding figures in the supplement) are not affected by this averaging?

- The paper is rich in details, sentitvity studies, and results which is highly appreciated. On the other hand it not easy for a reader to condense the wealth of findings from the almost 40 figures into a neat summary that reveals the main physics of the simulations. This seems feasible though:

  First, the inspection of the Figures 5, S15, S19, S23 a) suggests that the mean force scales with the Youngs modulus as $\overline{F} \sim E^{-\beta}$ with $\beta \approx 1/2$. The scaling with cohesion follows something like $\overline{F} \sim C^{\alpha}$ where $\alpha$ seems to be around $1 < \alpha < 3/2$. This is consistent for all snow types. Second, the inspection of Figures 2, S16, S20, S24 b) reveals that the slope $\lambda$ of the broken bonds percentage per unit length in the (bottom) linear regime correlates well with the initial contact density $\nu = z\phi$ (cf. [1]) when the volume fraction $\phi$ and the coordination number $z$ (evaluated through $z = N_{\mathrm{bonds}}/N_{\mathrm{clumps}}$) is taken from Tab 1. At the same time $\lambda$ is shown to be, in first approximation independent of the contact law parameters.

  So a simple law that is suggested by these observations, motivated by previous findings, and consistent with dimensional analysis would be

  $$\overline{F} = \mathrm{const}\, DC \left(\frac{C}{E}\right)^{1/2} \nu^{\gamma}$$

  where $D$ is the initial contact area from the contact law (Eq 1 in the paper) and an unknown exponent $\gamma$. If therefore the dimensionless variable $F\sqrt{E}D^{-1}C^{-3/2}$ is plotted against $\nu$, it would allow to merge all simulations for all parameters and snow types, including the sensitivity studies into a single figure at a glance while making contact to existing ideas for these elastic-brittle DEM simulations. Absence or presence of data collapse in this figure would greatly help to exploit the results, thus increasing the impact of the study.

Best regards,
Henning Löwe

**Minor comments**

(**Main text:**):

(l165): I don't understand "in the sense of the power diagram".

(l175): Table 1: Re-format such that units of density and ssa are not split.

(l175): As a quick cross-check from Table 1: Shouldn't be $n_{\mathrm{clumps}}d_{\mathrm{opt}}^3/\rho$ roughly equal to a constant ($\sim$ simulation container volume)? That does seems to work, but only RGlr is an outlier here. Why is that?

(l188): What does "weighting the bond magnitude between grains according to the spheres size" mean? The used $D$ values for each snow type should be included in one of the tables.

(l191): fails → fail

(l191): Explanation not clear, what is meant by "scale the normal stiffness in order that all the sphere-sphere interactions between two grains fails at the same moment"

(l196): Does this mean the cohesive contact fails only in normal direction/tension?

(195): Give $K_N$ here in terms of $E$ and radii here for completeness.

(l198): Unclear: "relative displacement..."

(l203): Sentence unclear.

(l229): Unclear, where is it applied?

(l233): And throughout: mix of italic and roman fonts for variables (like E, C, etc) in equations and in text.

(l265/266): Wouldn't it be way easier to interpret the values if the statistical descriptors were only evaluated for $z > 5$mm where the transients mostly died out? From fig 4 its obvious that in the upper half the statistics is different. Same in Fig S22b. This is stated somewhere later anyways.

(l265/266): Definition of the error metric unclear. Maybe an equation would be easier.

(l295): Fig 2, S6b, S8b, S10b: Could you please also add the non-averaged data for the broken bonds? This is helpful to document absence or presence of intermittency (i.e. bond failure "avalanches").

(Fig 8): Is it possible to put shaded regions as uncertainty?

(l460-466): This argumentation is a bit confusing: Why does the Youngs modulus in YADE does not represent the material? If this was the case, why would the similarity to the ice values then support the actual choice of parameters? MAybe explaining a bit better what the influence of the contact model/grain representation is would help.

(496): Rephrase sentence "The larger...."

(l577): Just asking, but isn't this kind of data management a bit too old-school?

(**Supplement:**):

(Table 1): Is the number of grains "67882" (3rd row from bottom in the DH section) really correct?

(Table 1): headings: Does the term "grains" used here have the meaning as the term "clumps" from Tab 1 in the main paper? If yes, make consistent. In fact, why are the numbers of clumps/spheres so different? Smaller containers for the sensitivity? Probably stated somewhere but I overread this.

**References**

[1] Gaume et al 2017, `https://doi.org/10.1103/PhysRevE.96.032914`

---

## Author Response (AR1)

We thank both reviewers, Richard Parsons and Henning Löwe, for their insightful comments that helped to improve the clarity and quality of the paper. Please note that a mistake in the calculation of the correlation length of the penetration profiles has been identified and corrected, modifying some results.

Our responses have been inserted in bold italics below the reviewers' comments. Line numbers refer to those of the revised versions of the manuscript and SM. Changes to the manuscript are highlighted in the track-changes file.

**RC1**: 'Comment on tc-2023-30', Richard Parsons, 20 Apr 2023

**Specific Comments:**

- There were a few sentences in the abstract which I found myself reading a few times to understand, they may benefit from rewording to make reading flow better:
  - 'The initial microstructure and ice properties fed the model, which can reproduce the exact same test numerically'
  - 'When the contact law is adjusted'…what from? This is the first mention of the contact law
    - It's already been stated that it reasonably reproduces the measured values – quantitively what is the difference between reasonable and good?
  - Last sentence of abstract – how? What is meant by 'better'?
  - Overall I think that the discussion and conclusion read very well, but I'm not sure that the abstract summarises the key points clearly.

***We have modified the abstract to address your concerns and clarify the main highlights of the paper.***

- Section 2.2.2 - A figure may be well suited to demonstrating the parameters of the interaction model – could be combined with Figure 1 for example.

***The representation of all the parameters of the interaction model in a single and comprehensive figure is challenging. Contact forces and displacements are particularly difficult to represent. The interaction model used for this study has already been described in the literature, where figures explaining the model can be found (Hagenmuller et al., 2015; Mede et al., 2020). We have therefore decided not to add a new figure but rather improve the clarity of Sect. 2.2.2 by reworking the text.***

- Figure 1 – The black lines used to represent cohesive interactions in the zoomed window are not easily visible. I wonder if using a different colour would make this more clear.

***We have modified the figure and represented cohesive interactions with white lines, improving the clarity.***

- Line 267 – what is the implication of choosing different weightings for the mean macroscopic force error and justification for choosing a factor of 2? i.e could different weightings result in selecting different combinations of mechanical parameters for the model comparison and is a better fit available?

***Firstly, note that we have identified an error in the calculation of the correlation length and corrected the value and description/interpretation in the article and the SM. This has resulted in selecting different mechanical parameters for some samples (RGlr, DH and PP) compared to the first version. The choice of the factor 2 was arbitrary. We wanted to give more weight to the mean force in the error calculation as we consider it the most important parameter to reproduce, especially compared to the correlation length which we have more difficulty fitting. In principle, changing this weighting can result in selecting different parameter combinations for adjusting the experimental measurements. The table below indicates the best-fitting parameter combinations obtained by not using the weighting factor:***

| Sample | E (Pa) | C (Pa) | tan($\varphi$) |
|---|---|---|---|
| RG | $1.0 \times 10^9$ | $5.0 \times 10^6$ | 0.2 |
| RGlr | $1.0 \times 10^9$ | $1.0 \times 10^6$ | 0.5 |
| DH | $1.0 \times 10^{10}$ | $5.0 \times 10^6$ | 0.2 |
| PP | $1.0 \times 10^9$ | $2.0 \times 10^6$ | 0.5 |

*Note that only the parameter combination obtained for the RGlr sample has changed compared to the case where the relative error is calculated with a factor of 2 on the force. Referring to Table S3, note that for the RGlr sample, the relative error on the obtained mean macroscopic force is much higher ($RE_F = 36.0\%$) when the total relative error is computed with equal weights for all the parameters, than when applying a weight of 2 to the mean force relative error ($RE_F = 5.5\%$) (Table 3). This supports our choice to apply a different weight to the mean force relative error to compute the total relative error.*

- With reference to line 303, it's not conclusive from the plots presented in S8 (and S18) that the depth hoar necessarily follows the same observed behaviour.

*We agree that from Figures S8 and S18 it is less evident that DH sample macroscopic force follows the same behaviour as the other samples. In particular, the decrease in slope to a nearly constant value is less visible. However, the average force profile curve (bold line Fig S8) shows the start of a slope decrease at a depth of around 8 mm, supporting our assertion that DH follows the same trend as the other samples and reaches a nearly constant force value but at a higher depth. To be more conservative, we have reworked the description of the macroscopic force profile for sample DH in the main document on lines 319-322 and in the SM on lines 130-131.*
*From Figure S18, we find that DH seems to follow similar behaviour as the other samples.*

- Wrt line 309, as discussed in S2.1.2 the depth hoar seems to differ slightly

*We agree that slope change between the first and second stages in the cumulative evolution of broken bonds is not as clear for the DH sample as for the other samples. Yet, the addition in the figure of the rate of cohesive bonds broken per unit depth does show a slight increase in the first 2-3 mm of penetration. We have modified the text accordingly in lines 328-330 and lines 140-142 of the SM.*

- Section 3.3 – in general, using percentages or factors rather than quantitative descriptions (eg 'slightly over/underestimated' / 'agree fairly well') to compare results is much more helpful in demonstrating the comparison. A mixture of these approaches is currently used.

*We thank the reviewer for this suggestion. We have systematically added percentages and quantitative values to support our description of the results; we have also added a new table (Table S2) indicating the values of the statistical indicators obtained for the experimental measurements.*

**Minor comments:**

- Line 50 – 'measure' – we should probably say we derive mechanical properties from cone penetration rather than measuring them

*The sentence on line 53 has been amended to reflect the reviewer's comment.*

- Line 61 – typo 'along on snow'

*The word 'along' has been deleted (line 64).*

- Line 80 – typo 'despite the NHPP'

*'Despite' has been deleted and the sentence reworked in lines 78-82.*

- Line 96 – typo 'failures mode'

*It has been replaced by 'failure modes' line 91.*

- Suggest the title of Section 2.1 is adjusted to indicate this methodology refers to defining measured values of the microstructure. I'd attribute 'experiments' to the wider task of comparing test data to modelled outputs.

*The title of Section 2.1 has been changed to "Experimental measurements".*

- Throughout the text, the term 'experimental' seems to be used to mean 'measured'

*Yes indeed, in the text the term 'experimental' refers to experimental measurements as opposed to numerical modelling.*

- Line 140 – resisting force applied to the cone not the rod.

*The resisting force corresponds to the sum of the forces exerted on the cone and the rod, in both the experiments and the numerical simulations. Note that the largest fraction of the force is exerted on the cone (see the example figure below, obtained for a numerical simulation with DH sample). The term penetrometer has been used instead of cone on line 132.*

[Figure]

- Line 126 – we later refer to the sample depths in terms of mm. May be best to change 2 cm to 20 mm for continuity.

*The size unit has been changed from cm to mm on lines 118, 214 and 540.*

● Table 1 – put units for density and SSA on a separate line so they're not split over 2 lines

*Table 1 layout has been reformatted so as not to split units over two rows.*

● Line 204 – typo 'clumps' to become 'clump'

*Done.*

● Table 2 – with reference to the Cohesion parameter default value (1.0 x 10^6), a value of 2.0 x 10^6 seems to have been fixed in sensitivity studies (eg caption of Figure 4, S12 etc), please confirm if default value is 1.0 or 2.0 x 10^6?

*You are right, there is a mistake in Table 2, the default value for the cohesion parameter is 2.0 x 10$^6$ Pa.*

● Figures 2, 4, 6 – Force / Depth profiles could do with adjusting the x axis, removing dead space to better display data and to make trends more observable.

*The x-axis of Fig. 2 has been adjusted to 1.2 N. The x-axis of Figs. S6, S8 and S10 have been adjusted as well. The x-axis of Fig.4 (and Figs. S14, S18 and S22) has been adjusted to better display data. For Fig. 6 we decided to keep the same x-axis value for all the snow types to emphasise differences between them in terms of absolute force values.*

● Line 347 – figure reference should be to fig 5?

*Indeed the reference should be Figure 5 at lines 362 and 364. The changes have been made.*

● Line 453 – I think the values stated for DZ obtained from CT scans refer to RG, RGlr and DH respectively, but without the PP samples would be helpful to restate these for clarity.

*The sentence in lines 473-475 has been modified to clarify the absence of CZ obtained from CT scans: "In comparison, the CZ derived from μCT scans extends radially up to 1.7R, 1.5R and 1.9R for the RG, RGlr and DH samples, respectively (no measurement for PP sample)."*

● Line 564 – typo – delete 'and'

*Corrected.*

**RC2**: 'Comment on tc-2023-30', Henning Löwe, 21 Jun 2023

**Main points:**

- What is the influence of averaging the simulation data on fixed 4µm windows? To fully understand the simulations it would be helpful to include also the raw (non-smoothed) force signals alongside with the 4µm-window versions that are finally used to evaluate the statistical descriptors and compared to the experiments. The main question here is: Can the authors confirm that the results reported in l345 (i.e. the "more complex" behavior of the force standard deviation and the correlation lengths (Fig 5b,5c and corresponding figures in the supplement) are not affected by this averaging?

*In practice, a force value is saved at each timestep during the simulations. Note that this timestep is not constant among the simulations, as it depends on the DEM stiffness parameter (see Eq. (6)). Accordingly, the corresponding depth-step ranges between $3.0 \times 10^{-9}$ m to $4.0 \times 10^{-8}$ m, depending on the simulation. We chose to average the numerical force data over a 4 µm window to match the SMP measurements and reproduce the experimental conditions as closely as possible. Below are comparisons between force profiles generated with raw (unaveraged) numerical data (left) and with the 4 µm averaging window (right) for each sample. It can be seen that the influence of the averaging is negligible on the overall shape of the force profiles. However, the averaging might have some influence on the amplitude of force fluctuations and the correlation length.*

*raw*                               *window = 4 µm*

*RG:*

[Figure]

[Figure]

*We show below plots of the three statistical indicators, namely the mean force, the amplitude of force fluctuations and the correlation length, as a function of the DEM mechanical parameters when computed from raw force data for each sample.*

[Figure]

*Comparison with the corresponding figures obtained from the averaged data (Figs. 5, S15, S19, S23) confirms that the mean force is not influenced by the averaging window (ratio (averaged/raw) of 1.001 between values from averaged data and raw). The values of the amplitude of force fluctuations are slightly decreased by the averaging window (ratio of 0.911 ± 0.026 between values from averaged data and raw). The averaging window has the most influence on the correlation length (ratio of 2.417 ± 0.775 between values of averaged and raw data). As also visible on the force profiles, the number of spikes tends to decrease, and the distance between two spikes tends to increase, in the averaged data compared to the raw data.*

*From these results, it can indeed be concluded that the averaging of the force data influences the resulting values of force fluctuations and correlation length, but this influence is limited and does not change the trends and conclusions made based on 4 µm window-averaged results.*

*We decided not to add the raw profiles to the paper (or to the SM) because their characteristics depend on the numerical parameters (time step), and we think that it is more meaningful to only*

*discuss the profile whose characteristics are the closest to the experimental profiles that we want to compare with our numerical results.*

- The paper is rich in details, sentitvity studies, and results which is highly appreciated. On the other hand it not easy for a reader to condense the wealth of findings from the almost 40 figures into a neat summary that reveals the main physics of the simulations. This seems feasible though:

    o First, the inspection of the Figures 5, S15, S19, S23 a) suggests that the mean force scales with the Youngs modulus as $F \sim E{-}\beta$ with $\beta \approx 1/2$. The scaling with cohesion follows something like $F \sim C\alpha$ where $\alpha$ seems to be around $1 < \alpha < 3/2$. This is consistent for all snow types.
    o Second, the inspection of Figures 2, S16, S20, S24 b) reveals that the slope $\lambda$ of the broken bonds percentage per unit length in the (bottom) linear regime correlates well with the initial contact density $v = z\phi$ (cf. [1]) when the volume fraction $\phi$ and the coordination number z (evaluated through z = Nbonds/Nclumps) is taken from Tab 1. At the same time $\lambda$ is shown to be, in first approximation independent of the contact law parameters.

So a simple law that is suggested by these observations, motivated by previous findings, and consistent with dimensional analysis would be

$$F = const \; DC \; (C/E)^{1/2}v^\gamma$$

where D is the initial contact area from the contact law (Eq 1 in the paper) and an unknown exponent $\gamma$. If therefore the dimensionless variable F √ED−1C−3/2 is

plotted against v, it would allow to merge all simulations for all parameters and snow types, including the sensitivity studies into a single figure at a glance while making contact to existing ideas for these elastic-brittle DEM simulations. Absence or presence of data collapse in this figure would greatly help to exploit the results, thus increasing the impact of the study.

*Thank you very much for this insightful analysis! We recognise that all the data/graphs presented in this document can be difficult to analyse and that a summary of the graphs/lists helps to summarise the results. Following your reasoning, we obtained the results below:*

- *Firstly, we computed exponents α and β of power laws derived from Figures 5, S15, S19, S23 for the Young's modulus and cohesion C respectively. Note α and β have been inverted compared to your comment.*

| sample | α (*F* vs E) | β (*F* vs C) | ω (λ vs E) | ξ (λ vs C) | ν (initial contact density) |
|---|---|---|---|---|---|
| RG | -0.48 | 1.37 | 0.01 | -0.04 | 0.55 |
| RGlr | -0.43 | 1.29 | 0.02 | -0.07 | 1.63 |
| DH | -0.43 | 1.35 | 0.04 | -0.07 | 0.86 |
| PP | -0.36 | 1.27 | 0.09 | -0.12 | 0.13 |

*Your estimations of α ≈ 1/2 and 1 < β < 3/2 are correct. Note that for PP sample, there is no value for E = 1 x 10¹⁰Pa, which might explain the slightly lower value of α and the higher value of ω. No*

consistent dependency of the mean macroscopic force with the friction coefficient for all the snow types could be retrieved from our results. The range of value tested for this parameter is narrow (between 0.2-0.5) compared to the ranges of the other two parameters and might prevent us from reliable calculation of the power law exponent. Therefore we have decided to present in the article the results obtained for a single friction coefficient (i.e. $tan(\varphi) = 0.3$). For the sake of transparency, we also present the results for all friction coefficients in our answer to reviewers.

- Secondly, we computed the initial contact density $\nu$ (above Table) and the slope of the proportion of the cohesive bonds broken $\lambda$, and obtained the following plot:

[Figure]

The slope $\lambda$ is mostly independent of Young's modulus (exponent of the power-law $\omega \approx 0$, above Table) and cohesion (exponent of the power-law $\xi \approx 0$, above Table) parameters. The initial contact density $\nu$ correlates well with the slope $\lambda$ for the physical parameters, following a linear-like trend.

- Thirdly, you can find below plots representing the data according to the suggested scaling law: $F\,T^{-1}\,E^{\frac{1}{2}}\,C^{-3/2}$ plotted versus $\nu$. Note that D in your proposed expression has been replaced by T as D is an already used symbol in the text (describe the sphere-sphere contact area). We explored different options for the T value.

    - First, we set T as the mean contact area between grains computed from segmented $\mu$CT images as suggested. The values are presented in the table below.

| Sample | Mean contact area of initial cohesive bonds (m²) |
|---|---|
| RG | $1.035 \times 10^{-8}$ |
| RGlr | $2.453 \times 10^{-8}$ |
| DH | $2.231 \times 10^{-8}$ |

| PP | 1.974 x 10⁻⁹ |
|---|---|

*The plot is displayed for all the friction coefficient values:*

[Figure]

*and for a friction coefficient of 0.3:*

[Figure]

*We observe no trend between the proposed law and the initial contact density. First, the values decrease to a contact density value between 0.5 and 1.0 and then increase.*

- *Second, we have set the T value as the tip area in contact with the sample (cone radius R = 2.5 mm and cone apex α = 60°):*

*The plot is displayed for all the friction coefficient values:*

[Figure]

*and for a friction coefficient of 0.3:*

[Figure]

*We observe that values obtained with the proposed scaling law for all the samples (initial contact density v) follow a logarithmic trend. This plot highlights the link between the microstructure main properties and the numerical mean macroscopic force. It attests to the ability of the contact law implemented into the model to reliably describe the physics at play between cohesive grains.*

*A new paragraph presenting the scaling law has been added to the manuscript (Section 4.2.4) and comments about these new results have been added to the abstract and conclusion.*

**Minor comments:**

- (l165): I don't understand "in the sense of the power diagram".

*The volume covered by a sphere and its relative power weight (proportional to the sphere's radius) was derived from a power diagram (or Weighted Voronoi Diagram) computed for spheres obtained using the medial axis method. The value obtained was used to filter out spheres whose covered volume is less than the minimum coverage defined by the S parameter. This allows non-essential spheres to be discarded to reduce the computation resources.*
*Modifications have been made to line 158 to clarify the explanation.*

- (l175): Table 1: Re-format such that units of density and ssa are not split.

*Table 1 layout was reformatted so as not to split units over 2 rows.*

- (l175): As a quick cross-check from Table 1: Shouldn't be nclumpsd$^3$opt/ρ roughly equal to a constant ($\sim$ simulation container volume)? That does seems to work, but only RGlr is an outlier here. Why is that?

| Sample | Bulk density (kg.m$^{-3}$) | SSA (m$^2$.kg$^{-1}$) | Number of grains | Optical diameter (m) = 6 / (SSA * ice density) | (Number of grains * Optical diameter$^3$ * ice density) / Bulk density |
|---|---|---|---|---|---|
| RG | 289 | 23 | 27560 | 2.84 x 10$^{-4}$ | 2.01 x 10$^{-6}$ |
| RGlr | 530 | 10.1 | 8488 | 6.48 x 10$^{-4}$ | 3.99 x 10$^{-6}$ |
| DH | 364 | 15.9 | 11211 | 4.12 x 10$^{-4}$ | 1.97 x 10$^{-6}$ |
| PP | 91.3 | 53.5 | 95022 | 1.22 x 10$^{-4}$ | 1.75 x 10$^{-6}$ |

*Indeed, the values obtained are of the order of 2.0 x 10$^{-6}$ m$^3$, except for the value of RGlr, which is roughly double (although of the same order of magnitude). The volume of the cylindrical container (r=1 cm and h=2 cm) is 6.28 x 10$^{-6}$ m$^3$, which is larger than all the volumes shown in the table above (the closest is the volume obtained for RGlr). The difference between RGlr and the other samples may have 2 causes:*

- *1) an under-segmentation or over-segmentation of the grains from the tomography scans, resulting in an increase or decrease in the number of grains. Segmentation involves a degree of subjectivity in the choice of segmentation parameters. It is more difficult to find satisfactory segmentation parameters for small, complex grains (PP, DH, RG) than for large, rounded grains (RGlr).*
- *2) Depending on the parameters (L and S) chosen to represent snow grains into discrete elements, a proportion of small grains are therefore removed in the numerical sample. There is then a mismatch between the number of grains in the numerical sample and the calculated SSA and bulk density (obtained directly from the tomography scans). This affects mainly samples with small grains such as PP, RG and DH. This is visible in Table S1 with the volumetric error Ev indicating the under-coverage of the DE sample compared to the provided segmented image.*

*From the previous fact, we presumably think that RGlr is fairly represented by the DE numerical sample while RG, DH and PP are presumably under-segmented and missing some small grains.*

- **(l188): What does "weighting the bond magnitude between grains according to the spheres size" mean? The used D values for each snow type should be included in one of the tables.**

*We wanted to reflect the fact that larger spheres are more likely to have larger contact areas, increasing the adhesion value and therefore the strength of the cohesive bond between the spheres. We feel that this sentence can be confusing and have decided to remove it. The whole paragraph has been reworked to improve clarity.*

**(l191): fails → fail**
*Done, replaced by 'break' in line 199.*

- **(l191): Explanation not clear, what is meant by "scale the normal stiffness in order that all the sphere-sphere interactions between two grains fails at the same moment"**

*Indeed, the definition of the contact parameters is quite subtle and specific to our model. The text has been modified as follows, intending to clarify the explanation (l. 191 - 204):*

*"The force of a given intergranular cohesive contact corresponds to the sum of all the associated sphere-sphere interactions. Based on the total contact surface between two grains (obtained from the µCT image) and the number of associated sphere-sphere interactions, each sphere-sphere interaction i can be associated with a representative contact surface $D_i$. In order to recover the correct cohesion strength between two grains, the adhesion parameter A was defined for each sphere-sphere interaction as:*

$$A_i = D_i\, C,$$

*with C (Pa) the cohesion of ice. In YADE, by default, the contact stiffnesses are computed based on the radii of the spheres in interaction and two elastic material parameters, namely the Young's modulus E and the Poisson ratio v. For our computations, to ensure that all cohesive sphere-sphere interactions between two grains break at the same separation distance, the computation of the normal stiffness was redefined as:*

$$K_{N,i} = \frac{D_i\, E}{r_{mean}},$$

*where $r_{mean}$ (m) is a characteristic length constant for all the interactions in the numerical sample, taken as the mean sphere radius. The shear stiffness is then defined as:*

$$K_S = v \times K_N\,."$$

- **(l196): Does this mean the cohesive contact fails only in normal direction/tension?**
*As now clearly stated on line 186, a cohesive contact can break either in normal (tension) or tangential (shear) directions, whichever threshold is first reached. When the bond breaks in tension, the contact between the two spheres is then lost. When the bond breaks in shear, the spheres may remain in contact with a purely frictional shear force.*
*We have modified the text in lines 186 to 190 to clarify our explanation.*

- **(195): Give KN here in terms of E and radii here for completeness.**
*To improve clarity, we have reorganised Section 2.2.2. The previous Eq. 4 that you are referring to is now Eq. 1. The definition of $K_N$ in terms of E and radii is now given in Eq. 4.*

- **(l198): Unclear: "relative displacement..."**
*We agree that this sentence was unclear and we have modified it on lines 183-184 to clarify that xs corresponds to the distance between 2 spheres in contact subjected to shear displacement.*

- (l203): Sentence unclear.

*This sentence was indeed unclear and has been modified on line 191. The idea was to explain how the contact force between grains (composed of several spheres) could be derived from the sphere-sphere contact law described above.*

- (l229): Unclear, where is it applied?

*The damping factor is applied to particle acceleration to dissipate kinetic energy and avoid numerical instabilities. The sentence on lines 233-234 has been modified to be clearer.*

- (l233): And throughout: mix of italic and roman fonts for variables (like E, C, etc) in equations and in text.

*The italic formatting has been generalised and the font has been changed to Times New Roman in the equations to be consistent with the text.*

- (l265/266): Wouldn't it be way easier to interpret the values if the statistical descriptors were only evaluated for z > 5mm where the transients mostly died out? From fig 4 its obvious that in the upper half the statistics is different. Same in Fig S22b. This is stated somewhere later anyways.

*The upper part of the SMP signal displays a transient behaviour for all samples in both the experimental measurements and the numerical results. We do not believe that it is problematic or complicated to use the whole profile to compute the statistical descriptors as, in this article, we aim to compare the experimental and numerical SMP signals to assess the capabilities of the numerical model to reproduce the experimental force profile. A later article could focus on the physical interpretation of the force signal itself, where computing descriptors in the steady-state part of the profile is indeed more relevant.*

- (l265/266): Definition of the error metric unclear. Maybe an equation would be easier.

*The definition of the metric has been replaced by equations (8) and (9) on lines 280 and 282.*

- (l295): Fig 2, S6b, S8b, S10b: Could you please also add the non-averaged data for the broken bonds? This is helpful to document absence or presence of intermittency (i.e. bond failure "avalanches").

*In the figures mentioned, we consider the cumulative number of broken bonds (in %) and not the averaged data as suggested in your comment. To follow your suggestion, we have added the rate of broken bond per unit depth on a second axis on Figs. 2b, S6b, S8b, S10b.*

- (Fig 8): Is it possible to put shaded regions as uncertainty?

*We have added the standard deviation of grain displacements as shaded regions for Figs. 8, 3, S7, S9 and S11.*

- (l460-466): This argumentation is a bit confusing: Why does the Youngs modulus in YADE does not represent the material? If this was the case, why would the similarity to the ice values then support the actual choice of parameters? MAybe explaining a bit better what the influence of the contact model/grain representation is would help.

*Discrete Elements (clumps of spheres) are used to represent snow grains. The contact between grains is approximated by this representation, contrary to Finite Element. The Young parameter implemented in YADE is a numerical parameter that locally represents the elastic properties of the material. Identity with the physical Young's modulus is not expected. The fact that our analysis supports that the value found is close to the real Young's modulus of the ice attests that the elastic properties are fairly well represented in YADE. But we agree the choice of Young modulus can be misleading. A sentence on lines 205-208 has been added to clarify this.*

- (496): Rephrase sentence "The larger...."

*The sentence in line 517 has been reworded as follows: "The mean macroscopic force, the amplitude of force fluctuations and the correlation length all increase with the cohesion C and, to a smaller extent, with the friction coefficient tan(φ)."*

- (l577): Just asking, but isn't this kind of data management a bit too old-school?

*We agree that, in the scope of open science, having all the data and codes stored on online platforms is an asset. However, this paper is based on a large amount of data acquired/obtained from various sources (SMP, tomography, DEM, experimental, numerical). Some of them require several processing steps (raw tomography scan, segmented images, DEM representation, article figures) and high-frequency output saving (DEM outputs). The volume of data is therefore significant (hundreds of GB). Storing this data online may not therefore be the best choice in terms of costs (price, volume, ecological) given the likely low demand for this specific data. If we are solicited, we will be very pleased to send the requested data/codes to the community.*

(Supplement:):

- (Table 1): Is the number of grains "67882" (3rd row from bottom in the DH section) really correct?

*Thank you for noticing this error. 67882 is the number of sphere-sphere interactions. The number of grains is 2527. It has been corrected in Table S1.*

- (Table 1): headings: Does the term "grains" used here have the meaning as the term "clumps" from Tab 1 in the main paper? If yes, make consistent. In fact, why are the numbers of clumps/ spheres so different? Smaller containers for the sensitivity? Probably stated somewhere but I overread this.

*Yes, the term 'grains' in Table S1 has the same meaning as the term 'clumps' in Table 1. We have made it consistent by choosing the term 'grains'.*

*The number of spheres is greater than the number of grains as we have chosen to represent grains with several spheres clumped together (cf Section 2.2.1). Depending on the choice of sphere characteristics (radius L and coverage S), the number of spheres required to approximate the grain shape varies.*

---

## Author Response (AR2)

Dear Melody,

We are pleased that our responses comply with the reviewer's comments. Thank you for your comments that helped to improve the clarity and quality of the paper. Our responses have been inserted in bold italics below your comments. Line numbers refer to those of the revised versions of the manuscript and SM. Changes to the manuscript are highlighted in the track-changes file.

- This is possibly only a technical correction but should the top half of equation 9 be log(measured) - log(computed)? Otherwise, this becomes problematic if you have a perfect simulation.
***You are right, the correct equation is (log(measured)-log(computed))/log(measured). This was an error in the text. The calculations have been done correctly. Thank you for pointing out this error, which has been corrected in Equation 9.***

- Please could you clarify how you derive the % errors from Tables 3 and 3S in lines 404-440. It is difficult to see how the values given in the text relate to the values in the table (also reference in line 439 should be to Table S3 rather than Table 3).
***The relative errors calculated with Equation 9 and presented in Tables 3 and S3 take into account the logarithm of the values. This makes it possible to select the best set of mechanical parameters, taking into account the difficulty of reproducing the correlation length and the fact that the statistical indicators vary by several orders of magnitude. To give the reader a better idea of the differences between measured and calculated values, the classical relative errors (without logarithmic values) are presented in the text (lines 404-440 and 435-457). We are aware that this can be misleading and that there is sometimes a mixture between these two values (for instance in the abstract). To clarify this, we explicitly mention the relative logarithmic error in the text when referring to the values presented in Tables 3 and S3. Otherwise, we refer to the relative error. Some values presented have been corrected accordingly, as have a few typos. We have also added sentences to lines 405-407 and 433-434.***
***Table S3 has been referenced instead of Table 3 in line 439.***